# USAR simulation system: spatial strategies for agent task allocation under uncertain conditions

Navid Hooshangi[1], Ali Asghar Alesheikh[2], Mahdi Panahi[3,4], Saro Lee[4,5]

[1] Assistant Professor, Department of Surveying Engineering, Arak University of Technology, Arak, Iran, Postal code 3818146763

[2] Professor in geospatial information science, Faculty of Geodesy and Geomatics Engineering, K.N. Toosi University of Technology, Tehran, Iran

[3] Division of Science Education, Kangwon National University, College of Education, # 4-301, Gangwondaehakgil, Chuncheon-si, Gangwon-do 24341, South Korea

[4] Geoscience Platform Division, Korea Institute of Geoscience and Mineral Resources (KIGAM), 124, Gwahakro Yuseong-gu, Daejeon 34132, Korea

[5] Department of Geophysical Exploration, Korea University of Science and Technology, 217 Gajeong-ro Yuseonggu, Daejeon 34113, Korea

**Correspondence:** Ali Asghar Alesheikh (alesheikh@kntu.ac.ir) and Saro Lee (leesaro@kigam.re.kr)

## ABSTRACT

Task allocation under uncertain conditions is a key problem for agents attempting to achieve harmony in disaster environments. This paper presents an agent-based simulation to investigate task allocation considering appropriate spatial strategies to manage uncertainty in urban search and rescue (USAR) operations. The proposed method is based on the contract net protocol (CNP) and implemented over five phases: ordering existing tasks considering intrinsic interval uncertainty, finding a coordinating agent, holding an auction, applying allocation strategies (four strategies), and implementing and observing the real environment. Applying allocation strategies is the main innovation of the method. The methodology was evaluated in Tehran's District 1 for 6.6, 6.9, and 7.2 magnitude earthquakes. The simulation began by calculating the numbers of injured individuals, which were 28,856, 73,195, and 111,463 people for each earthquake, respectively. Simulations were performed for each scenario for a variety of rescuers (1000, 1500, and 2000 rescuers). In comparison with the CNP, the standard duration of rescue operations with the proposed approach exhibited at least 13% improvement, with a maximal improvement of 21%. Interval uncertainty analysis and comparison of the proposed strategies showed that increased uncertainty led to increased rescue time for the CNP and strategies 1 to 4. The time increase was less with the uniform distribution strategy (strategy 4) than with the other strategies. The consideration of strategies in the task allocation process, especially spatial strategies, facilitated both optimization and increased flexibility of the allocation. It also improved conditions for fault tolerance and agent-based cooperation stability in the USAR simulation system.

**Keywords:** USAR operations; Agent-based simulation; Disaster Environments; Task allocation; Interval uncertainty; Spatial strategies.

## 1. Introduction

Preparation to manage an earthquake crisis requires optimal and appropriate management. Agent-based modeling of search and rescue operations after an earthquake is a good model for decision making, compared with traditional computational approaches [1]. Multi-agent systems consist of several automatic and autonomous agents that coordinate their activities to achieve a target [2, 3]. Multi-agent systems are suitable for the modeling and simulation of complex systems [4]. They allow the division of the system into subdivisions (agents) and the modelling of the relationships among these agents [5]. The use of multi-agent systems is necessary for disaster

management [6, 7]. Importantly, multi-agent systems can be used to implement various scenarios of search and rescue operations, as well as distributions of facilities, in the crisis area [2].

Task allocation is one of the main coordination challenges among sets of agents in a multi-agent system [8-10]. Agents fail to reach their ultimate goal without proper assignment of tasks [11]. In disaster environments, urban search and rescue (USAR) and the assignment of tasks are dynamic processes occurring under uncertain conditions [12]. Generally, task allocation on a large scale is influenced by uncertainties and various factors [13]. Uncertain conditions have a major impact on the initial planning and results of rescue operations [1]. Despite various investigations, an optimal task allocation solution has not been established [14].

In many instances, the initial allocation may result in problems or new tasks may be added to the worklist; therefore, reallocation is necessary. Reallocation is an effective reaction to environmental uncertainties and changes, and has important roles in both reducing the wasted time during an operation and increasing operation profitability [15]. Reallocation after instantaneous disruption is very important, especially in large-scale distributed systems (e.g., USAR operations) [14]. An effective task allocation approach in USAR operations should include strategies for replanning to manage future situations. Because tasks may not be performed well for various reasons, strategies such as minimum location displacement should be applied to initial responses to preserve additional time during reallocation or future task allocation. This approach to task allocation optimizes planning performance to achieve better performance time and provides conditions for fault tolerance.

The present article is the final part of a research project in Iran. This research project was carried out over three phases. In the first phase, uncertainty in task allocation among agents was considered and task allocation was performed only by considering the proximity (spatial distance) to the tasks. The developed method was evaluated in a square-shaped random environment without a sensitivity analysis [12]. In the second phase, the feasibility of the developed method was investigated in a simulated environment using real regional data. In this phase, the operational environment of a crisis was simulated and the developed method was examined in a real environment. In the simulated system, damage for a 6.8 magnitude earthquake damage was calculated for District 3 of Tehran, and rescue operations were modeled [1]. In the third phase using the concepts of previous articles [1, 12], spatial strategies were included in task allocation among agents and simulated with real-environment data. The present paper is the output of the third phase of the research project. The main purpose of the research is to improve task allocation in crisis-ridden conditions for agent-based groups by considering proper strategies to manage uncertainties. This paper first develops an agent-based simulation system for USAR operations, then applies uncertainties in agent decision-making by improving an interval VIKOR method to perform task allocation, and defines strategies for conditions under which the initial assignment has encountered a problem and requires reallocation (i.e., managing availability uncertainty during implementation). The innovation of the study is the establishment of an approach to improve conditions during reallocations or future allocations when initial allocations encounter problems due either to availability uncertainties or the addition of a new task. In general, spatial strategies are selected in such a manner that the final cost of the system will not increase abnormally if the initial allocations encounter problems. By applying spatial strategies in the assignment of tasks, it is expected that the tasks allocation in conditions of uncertainty will be done optimally and USAR operations will be performed more quickly.

## 2. Literature review and background

### 2.1 Agent-based USAR simulation

An agent-based model is a class of computational models for simulating the actions and interactions of autonomous agents. Agent-based simulations have been used in various investigations including crisis/disaster

management [1, 16], emergency supply chains [17], tsunamis [18], and collective behavior [19]. These simulations can be effective in both planning and policymaking [20]. Simulation of the operating system involves a simplified real environment, which is used to model a wide range of agents in complex systems. Various researchers have modeled a portion of the behavior of agents in simulated environments [16, 18, 21] and demonstrated collaboration among agents. However, agent cooperation in catastrophic environments has been less extensively studied, such that uncertainty in collaboration among agents has generally not been considered.

In previous studies, a geospatial information system (GIS) platform was used when preparing the environment and creating a simulation base map [19, 22, 23]. Spatial analysis and tools are used in most research endeavors in USAR operations [5]. Accurate and current spatial data play a vital role in earthquake-affected environments, such as assessing the extent of damage and casualties, surveying vulnerable areas after an earthquake, examining the existing infrastructure in the region, and finally tremendous aid in humanitarian efforts [2, 24]. Risk assessment of urban areas limits the impact of harmful events by increasing awareness of their potential consequences using spatiotemporal data [25]. Recommended tools in earthquake engineering sciences generally have a spatial basis and use spatial data and analysis [25]. Basic information, maps, and spatial tools in the form of Spatial Decision Support Systems (SDSS) and spatial frameworks such as webGIS have a significant impact on the speed of USAR operations [26]. Earthquake environment simulation is one of the important parts of agent-based modeling which is implemented using spatial analysis. To evaluate the vulnerability of buildings, some models and software based on infrastructures' spatial parameters have been developed [22], such as U.S. Geological Survey (USGS) model [27], HAZUS-MH (Multi-Hazard) [28], JICA[1] model [23], Federal Emergency Management Agency (FEMA) fragility curves [29], PO-ZID[2], and PO-AB[3] parametric methods [30]) and CIPCast-ES (Critical Infrastructure Protection - Earthquake Simulator) simulator [26].

## 2.2 Approaches to applying uncertainties in task allocation

Agents should include environmental uncertainties in their performance with respect to planning goals. There are four common approaches to considering uncertainty: probabilistic, fuzzy logic, rough set and interval set [12]. Uncertainty in task allocation has been investigated in various studies that can be categorized as sensor noise [21, 31, 32], an accidental event during execution [33, 34], the occurrence of new tasks [35, 36], the number of groups [37, 38], the relationship among agents [39, 40], and decision parameters [12].

The mentioned methods have been used in various applications such as multi- unmanned aerial vehicles [32], supply chains [38], moving plants [41], and disaster environments [40]. There is no dominant approach to model uncertainty for all phenomena. The appropriate method is determined based on the characteristics of the phenomenon and the purpose of the study. In crisis environments, there is uncertainty in all decision parameters. In the uncertainty in decision parameters category, which is suitable for multi-agent systems, uncertainties are associated with the decision parameters for assigning tasks. Therefore, all information needed for task allocation is considered uncertain. Various methods such as the contract net protocol (CNP) [12], stochastic scheduling [41], and genetic algorithms [42] have been used in these contexts. Here, we present an approach that includes uncertainties in decision parameters and strategies in the CNP. The CNP produces local optimal solutions that are abundantly used in multi-agent systems [39]. This method is simple, practical, and popularly used in agent-based modeling. In USAR operations, complete individual expertise is impossible due to a lack of environmental

---

[1] Japan International Cooperative Agency (JICA) model

[2] Potresne Odpornosti Zidanih (Slovenian language)

[3] Potresne Odpornosti armiranobetonskih konstrukcij (Slovenian language)

knowledge; therefore, determining membership function and the probability distribution is a complex and time-consuming step. We used interval analysis to manage these shortcomings and to consider the intervallic nature of available data within a rescue operation. In the interval set method, due to the uncertainty in a parameter's value, that parameter is specified as an interval regardless of the probabilistic distribution (unlike in probabilistic theory) or membership function (unlike in fuzzy logic) [12].

### 2.3 Reallocation and reassigning methods

Distinct algorithms have been proposed for scheduling and task reallocation in accordance with the tasks and available conditions within an environment [43]. Some reallocation methods (e.g., data envelopment analysis [44]) and exact algorithms (e.g., a branch-and-bound algorithm with column generation) resolve problems on a smaller scale (e.g., 10 jobs and three vehicles). In such methods, the process is time-consuming and slow for resolving large-scale problems [13]. Therefore, they are not suitable for the allocation of tasks that should be performed dynamically and instantaneously in large-scale problems.

In some research, such as the investigation of gate reassignment problems, initial assignment tables have been created using heuristic methods in such a manner that a succession delay is minimized [45]. The incidence of adverse events may disrupt the original table. Notably, this method is not suitable for a large number of tasks. Some other task allocation methods are interdependent with the plan's ongoing tasks, such as in construction operations [14]. In those mathematical calculations, when a task fails, all other tasks that were based on its correct implementation must be replanned.

An appropriate reallocation method must be applied with respect to the nature and scale of the problem. In USAR, a rescue process generally occurs independently of any other rescue processes, and only a portion of the workflow is ready to be implemented and assigned. Moreover, because of the large number of rescue groups in USAR operations, as well as the available uncertainties and dynamic nature of multi-agent systems in disaster environments, the concept of general planning is uncommon and appropriate plans should be produced both locally and cross-sectionally. Most available methods to resolve the problem of assigning tasks cannot be developed for uncertain conditions and restrictions such as in critical rescue environments (e.g., USAR after earthquakes).

With respect to USAR operations, task allocation methods must include different strategies for all conditions and be dynamically generated in a real-time environment. In contrast to previous studies, we define an approach based on spatial strategies, such that the results of the initial task allocation are used for future task allocations and are appropriate in the rescue environment. Time limitations constitute another issue in the reallocation and reassignment of rescue teams. Therefore, the present study aims to expand the CNP method for rapid problem resolution.

## 3. Case study and data

The proposed approach can be implemented in various study areas. This study used a part of Tehran (District 1 in the capital of Iran) to evaluate the feasibility of the proposed method on the basis of available data. District 1 is one of 22 central districts of Tehran Province, Iran. District 1 has an area of 210 km$^2$ and is located in the northernmost part of the city of Tehran (Figure 1). According to the 2016 census, its population is 433,500 people.

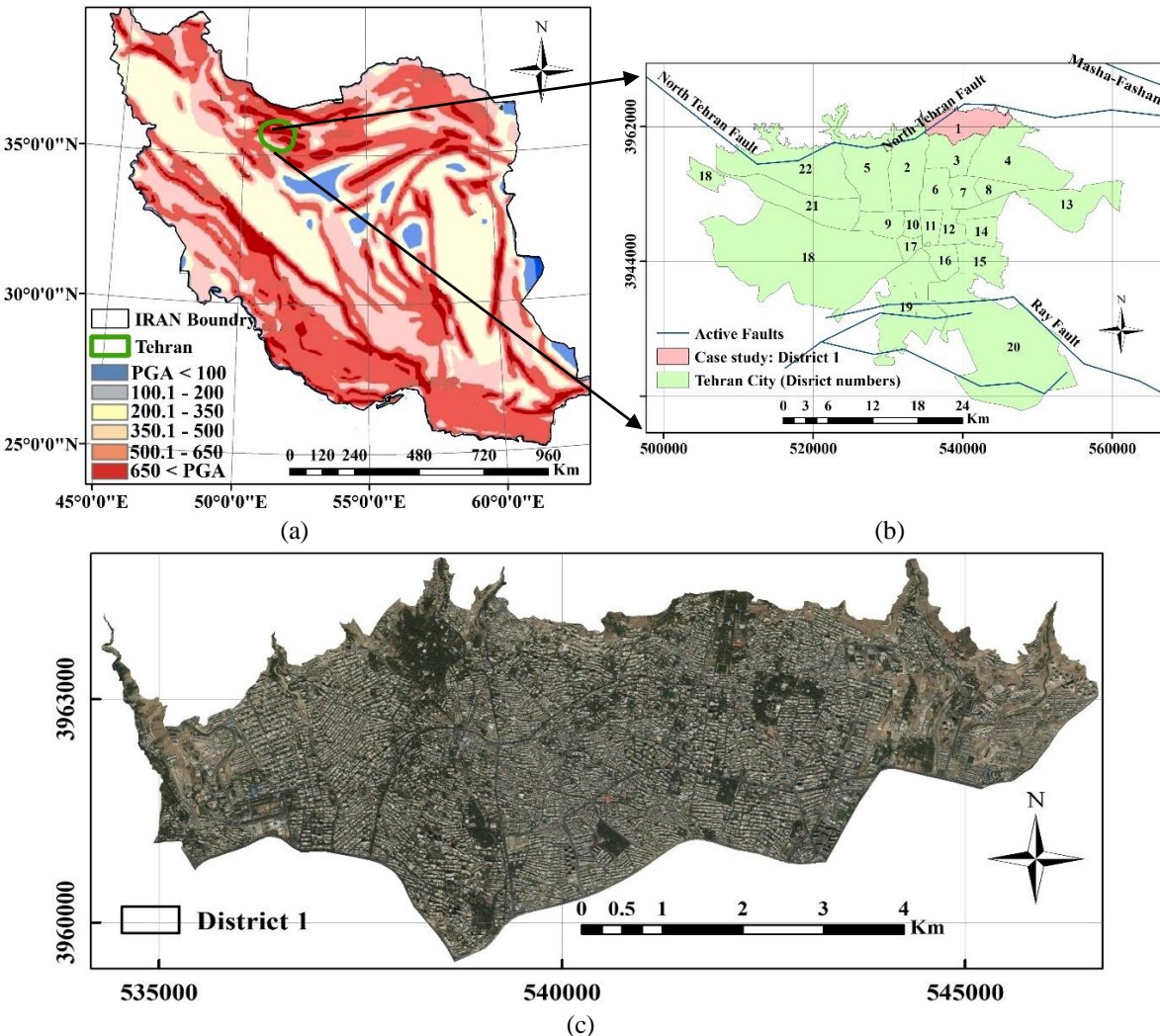

**Figure 1** Location of case study: (a) peak ground acceleration (PGA) map of Iran for a return period of 2475 years and approximate location of Tehran, (b) location of District 1 and active faults in Tehran (c) Map of District 1 (study area) and active faults, Tehran.

The recent Tehran earthquake (5.2 magnitude) in December 2017 attracted the attention of many urban
planning organizations. Tehran is a highly seismic area because it is surrounded by the Ray, Masha-Fasham, and North Tehran faults (Figure 1(b)) [46]. Tehran is located in the southern part of the Alborz Mountains, where a magnitude 7.3 earthquake occurred in 1990 [47, 48]. Tehran faults show some M7+ historical earthquake records [22, 46]. Seismologists have reported that Tehran is vulnerable to earthquakes and is expecting a destructive earthquake in the future [23, 49]. The North Tehran fault (NTF) is the city's largest and most prominent active
tectonic structure fault, which is approximately 175 km long [22, 50]. The paleoearthquakes study on this fault has revealed seven surface-rupturing events with magnitudes between 6.1 and 7.2 [22, 46, 51]. For this purpose, the North Tehran fault scenario, with the capacity to cause the most destructive potential earthquake in Tehran, was selected in the present study. The method developed in this research can be implemented for any scenario. In accordance with the previous earthquakes and suggestions of seismologist experts, we simulated 6.6, 6.9, and 7.2
magnitude earthquakes. The basic data used in environment simulation were block maps, population, distance from the fault, building material, agent location, year of building construction, and building height.

## 4. Materials and Methods

In this section, the simulated scenario and proposed method are described.

### 4.1 Scenario of proposed agent-based USAR simulation

We assume the presence of a disaster environment in which events are uncertain, for example, the time it takes to go from location A to location B is not exactly known. The injured individuals are trapped under rubble and the number of such individuals in each building block is uncertain. Rescuing injured people is the main goal. Saving each person is a task that must be performed through the cooperation of rescue agents. After an earthquake, the numbers of injured and deceased people can be estimated by using different formulas by determining the magnitude and location of the earthquake, as well as the urban context data of the buildings [52]. Furthermore, the possible locations of injured individuals can be predicted using building damage assessment models. Therefore, the simulation inputs are the injured individuals' locations and their characteristics, which are available with some uncertainty. The rescue agents are attempting to save injured individuals by moving toward to the task location. Given the results of previous studies [12, 42, 53, 54] and in accordance with expert opinion on USAR operations, the uncertainties include the number of injuries, severity of the victims' injuries, duration of the operation, infrastructure priorities, agent energy, route status, task runtime by an agent, and risk level for each agent. These are important uncertainties in task allocation. All parameters are specified as intervals during the task allocation process. After task identification, an agent is assigned a task and pursues it. If an agent fails to complete an assigned task because of any existing disruptions, the task is updated with respect to uncertainties and reported to the central agent, resulting in the re-initiation of the task allocation process. In this process, task allocation strategies are applied to minimize the cost of the system.

In this scenario, there is a central agent, as well as several coordinators, rescuers, and injured agents in the environment. These independent agents are rational and can communicate with each other. The agents have the following roles and characteristics:

- **Central agent:** This agent is responsible for sorting the tasks, specifying the coordinators, determining the results, announcing rescuers, and applying allocation strategies.
- **Coordinating agent:** The coordinator is a rescue agent who is responsible for sending work details to rescuers, receiving their proposals (bids), holding auctions, and submitting the results and rescuer prioritization data to the central agent.
- **Rescue agent:** This agent identifies and moves to the task location, searches for injured individuals, sends the task uncertainty to the central agent, and rescues injured individuals from the debris.
- **Injured agent:** This agent exists in the environment and has a critical condition that changes continuously. This agent has no activity or communication with other agents.

### 4.2 USAR simulation

In preparation for the USAR operation simulation, there are three main parts: 1) calculating the damage rate of the area and people (simulating an earthquake-damaged environment), 2) defining agents and their characteristics, and 3) implementing the suggested method for task allocation between agents.

To simulate an earthquake-damaged environment, an earthquake risk assessment model was developed based upon the Japan International Cooperative Agency (JICA) model. The JICA model is the output of cooperation between the Center for Earthquake and Environmental Studies of Tehran and the JICA. The results of this project and its implementation have been presented previously [55] and used in various studies [1, 23, 48]. This model can calculate the buildings' level of destruction and the number of injured people based on the earthquake intensity, earthquake location, building vulnerability, and the population in these buildings [48]. The steps for creating a vulnerability map and finding casualties based on the JICA model are shown in Figure 2.

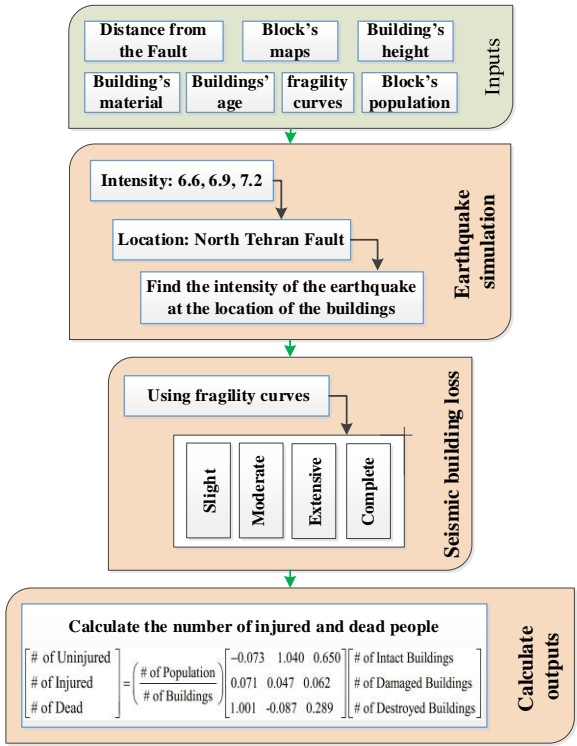

**Figure 2** Steps for building damage assessment and calculating the number of injured people [23, 24, 55].

AnyLogic software is used to simulate the scenario described in the previous section based on multi-agent systems. In our scenario, we included four types of agents: injured individual, rescuer, coordinator, and central agent. In AnyLogic software, a statechart is designed to suit the tasks of each agent. Statechart determines the work process of the agent. The tasks described in the previous section were implemented for each agent. The simulated agents in the environment are independent, located in a specific place, logical and decision-making, can move to a specific location, and other agents, except the injured agent, communicate with each other. In simulating USAR operations, location and the use of maps and spatial analysis play a key role. AnyLogic software can process geospatial information system data. The initial locations of injured agents were based on building damage and the locations of rescue groups were randomly generated in the environment. The definitions of agents and their characteristics were described in detail in our previous article [1]. The relevant agents move along the central line of the road and use the Dijkstra algorithm to find the shortest path. Dijkstra's algorithm is a well-known algorithm for finding the shortest paths in road networks [5].

There are many injuries in the environment. The rescue of the injuries is possible with the cooperation of the agents. The process of cooperation between agents is simply put as follows: The central agent first sorts the tasks according to their priorities. After the coordinating agent has been determined, the central agent sends the task properties to the coordinating agent. The coordinator holds an auction. Rescue agents bid in accordance with their environmental and working conditions. Rescuers are in a ready state at the start of the operation. Each successful rescue agent moves to the task's location. After reaching the task position, the rescue agent begins rescuing the injured agents. During the execution of their assigned work, the agents may find considerable differences between the real-world information and the information expressed in the auction. In such instances, the agents may stop performing their tasks and report the discrepancies to the central agent. The method of cooperation between the agents is described based on the proposed method in the next section.

### 4.3 The proposed agent-based task allocation method

The proposed model for task allocation with uncertainties in earthquake USAR operation is shown in Figure 3.

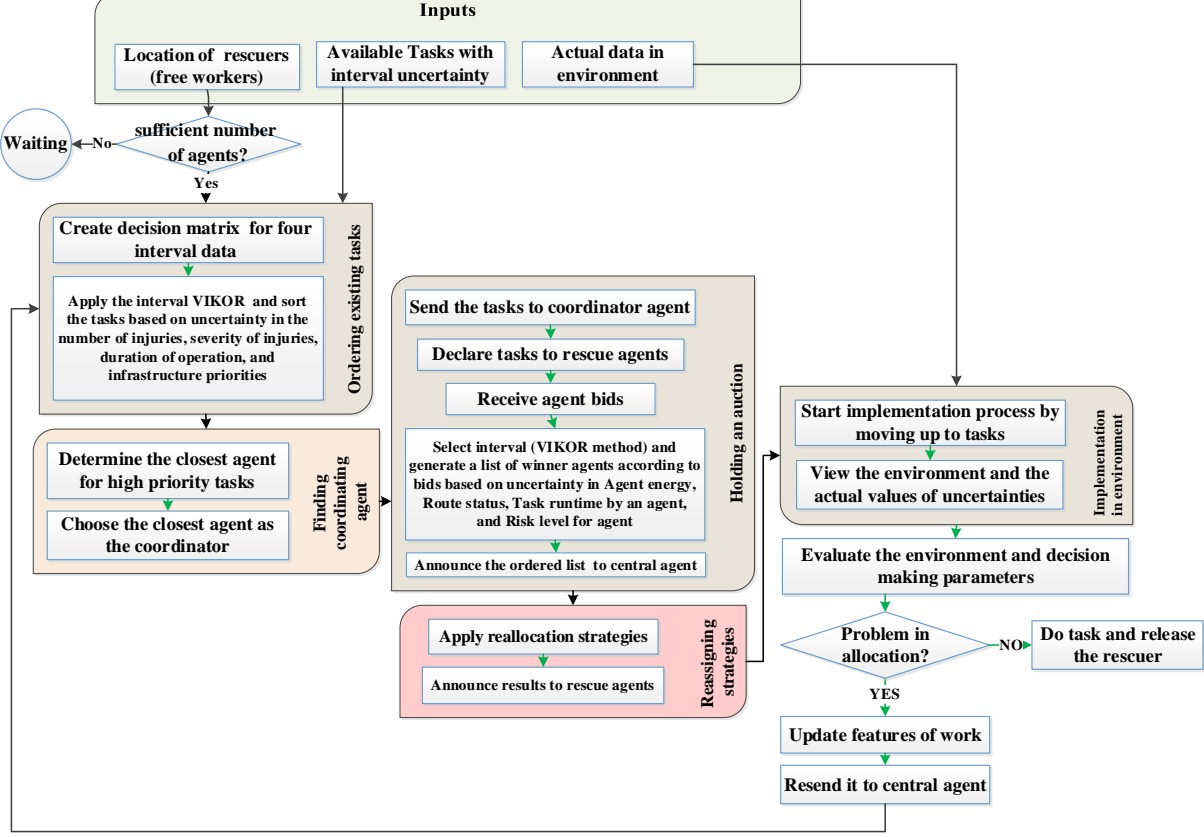

**Figure 3** Task allocation flowchart in the proposed approach, separated into five steps within an environmental simulation

The five steps of the proposed approach are the ordering of existing work, specifying the coordinators, holding an auction, applying reassignment strategies (the innovation of this paper), and implementing and observing environmental uncertainties (performed by an agent). The proposed method is presented below.

#### 4.3.1 Ordering existing tasks

In crisis-ridden areas, there are varying degrees of urgency [54]. Tasks with a higher priority must be performed first. Four parameters are used to prioritize tasks: the number of victims, severity of injuries, time required for a rescue operation, and infrastructure priorities. The initial tasks with their uncertainties in the environment (four priority parameters) are available to the central agent. Therefore, for each task feature, an interval such as that expressed in Table 1 is specified.

**Table 1** Task characteristics based on intervals

| Task no. | X coordinate | Y coordinate | Infrastructure priorities | Number of injuries | Severity of victim injuries | Duration of operation |
|---|---|---|---|---|---|---|
| 1 | $X_1$ | $Y_1$ | $[I_{l1}, I_{u1}]$ | $[N_{l1}, N_{u1}]$ | $[S_{l1}, S_{u1}]$ | $[D_{l1}, D_{u1}]$ |
| 2 | $X_2$ | $Y_2$ | $[I_{l2}, I_{u2}]$ | $[N_{l2}, N_{u2}]$ | $[S_{l2}, S_{u2}]$ | $[D_{l2}, D_{u2}]$ |
| ... | ... | ... | ... | ... | ... | ... |
| i | $X_i$ | $Y_i$ | $[I_{li}, I_{ui}]$ | $[N_{li}, N_{ui}]$ | $[S_{li}, S_{ui}]$ | $[D_{li}, D_{ui}]$ |
| ... | ... | ... | ... | ... | ... | ... |
| n | $X_n$ | $Y_n$ | $[I_{ln}, I_{un}]$ | $[N_{ln}, N_{un}]$ | $[S_{ln}, S_{un}]$ | $[D_{ln}, D_{un}]$ |

At this stage, all MCDM decision methods such as VIKOR[4], TOPSIS[5], ELECTRE[6], etc. can be used. In previous researches, the predominant method in this field has not been recommended. On the other hand, since we use interval instead of a number in multi-criteria decision making, we use MCDM methods that can consider an interval instead of a number. In previous studies, TOPSIS and VIKOR have been used based on interval values, but no particular superiority has been observed [1, 12, 56]. In this research, the interval VIKOR method is used to sort tasks and compare the agents' bids. The VIKOR method is done in five main steps [56, 57]: First, the decision matrix with interval data is formed so that the rows represent the alternatives (A), the columns represent the criteria (C), and the matrix values ($f_{ij}$) represent the value of alternatives i relative to criterion j. Matrix values are interval ($f_{ij} \in [f_{ij}^L, f_{ij}^U]$). Then the positive (PIS) and negative ideal solution (NIS) is determined. The positive ideal solution is the highest column value for the profit criterion and the lowest column value for the cost criterion. Then S and R intervals are calculated and based on them, the interval Q is calculated using Equations 1.

$$S_i^L = \sum_{j \in I} w_j \left( \frac{f_j^* - f_{ij}^U}{f_j^* - f_j^-} \right) + \sum_{j \in J} w_j \left( \frac{f_{ij}^L - f_j^*}{f_j^- - f_j^*} \right) \qquad i=1, \ldots, m$$

$$S_i^U = \sum_{j \in I} w_j \left( \frac{f_j^* - f_{ij}^L}{f_j^* - f_j^-} \right) + \sum_{j \in J} w_j \left( \frac{f_{ij}^U - f_j^*}{f_j^- - f_j^*} \right) \qquad i=1, \ldots, m$$

$$R_i^L = max \left\{ w_j \left( \frac{f_j^* - f_{ij}^U}{f_j^* - f_j^-} \right) \bigg| j \in I, \quad w_j \left( \frac{f_{ij}^L - f_j^*}{f_j^- - f_j^*} \right) \bigg| j \in J \right\} \qquad i=1, \ldots, m$$

$$R_i^U = max \left\{ w_j \left( \frac{f_j^* - f_{ij}^L}{f_j^* - f_j^-} \right) \bigg| j \in I, \quad w_j \left( \frac{f_{ij}^U - f_j^*}{f_j^- - f_j^*} \right) \bigg| j \in J \right\} \qquad i=1, \ldots, m$$

$$Q_i^L = v \frac{S_i^L - S^*}{S^- - S^*} + (1 - v) \frac{R_i^L - R^*}{R^- - R^*}$$

$$Q_i^U = v \frac{S_i^U - S^*}{S^- - S^*} + (1 - v) \frac{R_i^U - R^*}{R^- - R^*}$$

$$S^* = \min_i S_i^L, \quad S^- = \max_i S_i^U, \quad R^* = \min_i R_i^L, \quad R^- = \max_i R_i^U$$

In the above formula, A* and A⁻ are PIS and NIS, i is associated with benefit criteria, and J is associated with cost criteria, wj is the weight of criterion Cj and v is introduced as the weight of the strategy of ''the majority of criteria'' (or ''the maximum group utility''), here suppose that v = 0.5. Finally, find an appropriate alternative based on Q Intervals. A better option is to have a smaller Q interval than the others. The following constraints are used to calculate the smaller interval. If a=[$a^L$ $a^U$] and b=[$b^L$ $b^U$], the comparison between these two intervals is as follows:

- If the two intervals do not have an intersection, the interval whose values are lower is minimum interval number.
- If the values of the two intervals are $a^L \leq b^L < b^U \leq a^U$, the interval a is minimum if $\alpha(b^L - a^L) \geq (1 - \alpha)(a^U - b^U)$.
- If the values of the two intervals are $a^L < b^L < a^U < b^U$, the interval a is minimum if $\alpha(b^L - a^L) \geq (1 - \alpha)(b^U - a^U)$.

### 4.3.2 Finding the coordinating agent

---

[4] Vlsekriterijumska Optimizacija I Kompromisno Resenje

[5] Technique for Order of Preference by Similarity to Ideal Solution

[6] ÉLimination et Choix Traduisant la REalité

For each task defined by the central agent, the most appropriate agent is identified as the coordinating agent. The coordinating agent is an agent who is located near that task and is not currently working. The selection of a coordinating agent and creating groups to execute any task can be achieved through different methods and is based on various criteria [10, 58]. In this study, to simplify the calculations, only the criterion of proximity (spatial distance) is used to identify the coordinating agent. Therefore, the nearest agent to the task is selected as the coordinator and is responsible for the auction. Selection of a the coordinating agent leads to the performance of calculations at a distributed point. By selecting coordinating agents, the computational overhead of the central agent is reduced.

### 4.3.3    Holding an auction

Coordinating agents hold auctions after receiving the task characteristics and the list of agents in the subgroup. In the CNP, agents bid for tasks, and the agent who offers the highest value for the task is the winner. During the auction, rescue agents offer intervals (rather than values) for the route conditions, the time required for the agent to execute the task, the agent's possible risk level, and their energy. Accordingly, the agent calculates numbers for each of the four decision-making criteria, such as variable X, based on Equation 2. In Equation 2, the distance (in meters), severity of the victims' injuries, and task priority are based on values declared by the central agent. Based on the rate of uncertainty presumed for a given environment (for example, 30%), an interval for this number is estimated. The first number of this interval is in the range between $[X, X + 30\%X]$ and the second number is in the range $[X - 30\%X, X]$.

$$
\begin{aligned}
&\text{Agent energy (energy level, distance, number of people)} = \text{Energy level} - \text{Distance}/500 - \text{Number of people rescued}*0.3 \\
&\text{Task runtime by an agent (distance, number of people, severity)} = \text{Distance}/150 + \text{Number of people rescued} *15 + \text{Severity}*2 \\
&\text{Risk level for an agent (energy level, priority)} = \text{Priority} - \text{Energy Level} \\
&\text{Route status (distance)} = \text{Distance}
\end{aligned} \tag{2}
$$

In the real world, each person can introduce intervals according to their experience and their knowledge of the environment. In this study, we used the above equations based on expert opinion to simulate the real environment. In some previous researches, special formulas have been used to calculate the agents' scores based on the difficulty of the task, the priority of the task, the duration of each activity, and so on [14]. The use of such a formula is not very appropriate due to the fact that it strongly affects the results of the action and is practically depends on expert opinions. In this step, instead of using the formula to evaluate the agents' bid, the interval VIKOR method is used to find better propose. Then, the coordinating agent applies the interval-based VIKOR method to order the agents' bids. The coordinating agent sends the results to the central agent after ordering the agents. The use of a central agent in this phase provides the opportunity to make the best decision considering the task priorities and capacities of other agents.

### 4.3.4    Applying allocation strategies

In operations where there is uncertainty, the issue of task allocation cannot be definitively resolved. In this phase, the initial allocation should be done in such a manner that a potential reallocation would waste the smallest amount of time. Based on different strategies at this stage, the central agent begins to assign tasks after obtaining all lists from coordinating agents. In each strategy, a priority is assigned to specific tasks. In this section, four different strategy-based approaches are described, as follows:

***Task allocation according to priority (strategy 1):*** In this strategy, task allocation begins with the assignment of higher-priority tasks, following establishment of the task order and priorities of the rescue team in the previous stage (prioritization and auction). Therefore, the agent with the best performance is selected for high priority tasks

and is subsequently excluded from the lists of agents with no tasks. Subsequently, the tasks of lower priority are assigned in the same order. The limitation of this strategy is that it may cause some agents to not receive tasks.

***Assigning tasks to all agents, preferably to specific agents with optimal outcomes (strategy 2):*** This strategy is based on the optimal use of all rescue teams. In this strategy, all agents are assigned a task. For this purpose, a task is first assigned to an agent who has applied for the minimum number of tasks. The agent and task are then

eliminated from the agent and task lists, and the allocation continues with the next agent who has made few requests. Using this strategy, a task will be assigned to all agents.

***Task allocation on a strategic spatial basis (strategy 3):*** Using on this strategy, agents who play important and strategic roles in the task allocation process are excluded to ensure their availability for the implementation of tasks if problems are encountered during the task allocation process. Agents with strategic roles may be defined

differently. Agents who participate in the auctions of more tasks are those with strategic locations. In such instances, these agents are close to many tasks (have strategic spatial locations) and can be used when these tasks are not implemented. Figure 4 shows the difference between the task allocation results for strategies 2 and 3. In Figure 4, a rescue agent located centrally has a strategic position and will try to maintain this position. Although the total movement may increase, if there are problems in performing other tasks, this agent can help all other

groups.

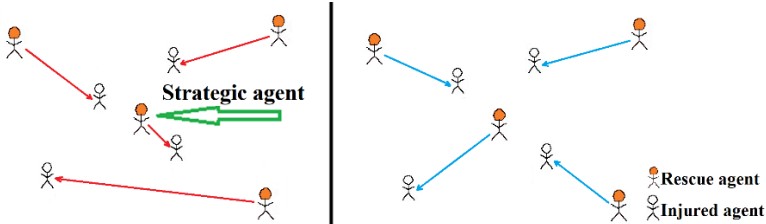

**Figure 4** Strategic agent illustration. Blue arrows show the final results for strategy 2 and red arrows show the successful rescuers in strategy 3.

***Assigning tasks by creating the best density in the environment (strategy 4):*** This strategy is based on the optimal density of rescue agents. Using this strategy, task assignments are made in a manner that ensures the uniform distribution of agents in the environment. Generally, no exact information is available concerning the

335 conditions of the tasks; therefore, this strategy aims to ensure a uniform distribution of rescue teams within the environment if the uncertainty is high. In disaster environments such as earthquakes, the incident occurs over a wide area, such that the damage and injured population are uniformly distributed due to the texture of the area. Therefore, the highest number of injured people is not accumulated in any one spot. Furthermore, applying this strategy prevents the convergence of rescue teams. To apply this strategy, the tasks of the highest priority in the

340 task lists should be given to the available agents where the environmental density is the highest. The concept of optimal density can be solved through innovative algorithms. In our study, the simulated annealing method was used to determine uniform density. The implementation stages of simulated annealing have been described previously [3]. Figure 5 shows the difference between task allocation outcomes for strategy 2 and strategy 4.

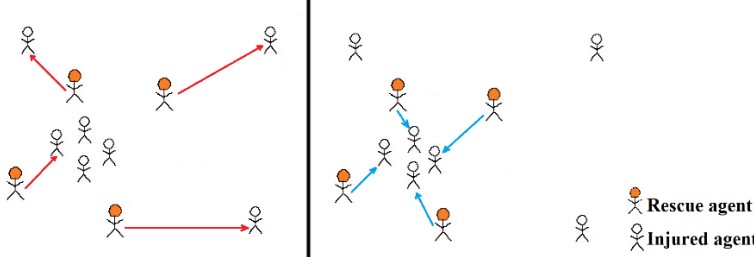

**Figure 5** Best density strategy illustration. Blue arrows indicate the successful rescuers in strategy 2 and red arrows indicate the final results for strategy 4.

### 4.3.5 Implementation and observation of real values in the environment

During the implementation phase, tasks are implemented by agents in a dynamic environment where there are always uncertainties during task execution. The rescuer observes the difference between predicted values and the actual environment after the work begins. In this study, a random number in the $[X - 30\%X, X + 30\%X]$ interval was chosen to model the real environment. In the real world, the difference between the predicted environment (through building vulnerability estimation models) and the real environment will determine the agent's
performance.

If the agent observes a large difference between the auction information and the real environment, the agent abandons that task. In this instance, the agent updates the task's values and uncertainties and returns the work to the central agent. The new uncertainty interval will be 80% smaller than the original interval. There are various conditions under which agents will reallocate a task if the environment differs from the expected scenario. For
example, the agent can abandon the task if three of eight decision-making parameters are out of range by 5%. Otherwise, the agent finishes the rescue work by accepting the new conditions.

The central agent assigns newly added tasks within the reallocation framework. When a new task is assigned, the task allocation is combined with that of both new and incomplete tasks.

## 4.4 Evaluation method

Assessment of a task allocation algorithm is typically performed in the first phase through modeling and simulation due to the dynamic and heterogeneous nature of different environments [14]. Simulation is a suitable approach for the implementation and validation of a proposed method [9]. In a real test situation, the situations and conditions of the implementation scenario are difficult to reproduce. In the present study, we simulated three scenarios for earthquakes in Tehran's District 1 with magnitudes of 6.6, 6.9, and 7.2. We also estimated the
numbers of deceased and injured individuals who are distributed in the centers of relevant building blocks and need to be rescued by 1000, 1500, or 2000 rescue agents. In the uncertainty analysis of the suggested method, the lower and upper bounds of uncertain values were also calculated. The proposed method was compared with the traditional CNP. The intended task allocation was considered efficient if profitability parameters were maximized. In accordance with to several recent studies [8, 12, 53], three criteria were used to evaluate the performance of
the proposed method: the number of deceased victims, number of incorrect allocations, and rescue time. Results

Some of the major problems in reallocation and in the task allocation environment include scalability, reliability, performance, and dynamic resource reallocation [43]. In this study, the results of two analyses (scalability of the proposed method and interval uncertainty analysis) are presented.

The first analysis focused on the evaluation of the proposed approach at different scales and for different
criteria. Comparison and assessment were carried out at different scales to measure the effectiveness of the proposed approaches in USAR operations. Nine scenarios were applied in this study and compared with traditional the CNP.

The second analysis focused on interval uncertainty analysis and studied the rescue operation duration in the 6.9 magnitude earthquake at different levels of uncertainty. In this analysis, time changes in rescue operations were investigated according to different levels of uncertainties. The duration of a rescue operation in the simulation model depended on two main components: prioritization of tasks and, outputs of each operation in each phase [1]. Equation 3 defines the final model for calculating the operation duration based on these two components.

$$T(x_1, x_2, x_3, x_4, x_5, x_6, x_7, x_8) = \sum_{n=1}^{n+1} \alpha_n(x_1, x_2, x_3, x_4) + \sum_{w=t}^{n+t} \beta_w(x_5, x_6, x_7, x_8) \qquad (3)$$

Variables x1 to x8 constitute the number of injuries, severity of injuries, duration of the operation, infrastructure priorities, energy, route status, task runtime by agents, and risk level for agents, respectively. $\alpha_n$ is the function of task prioritization and $\beta_w$ is the function of bidding.

To our knowledge, interval uncertainty analysis has rarely been employed. The method used in this research was adapted from previous literature [59]. In our analysis, Chebyshev points are used. Equation 4 depicts a Chebyshev formula for generating m collocation points in the interval [0, 1] [59]:

$$number_i = \begin{cases} 0.5 \times \left[1 - \cos\left(\frac{\pi(i-1)}{m-1}\right)\right] & for \ j = 1, if \ m = 1 \\ 0.5 & for \ j = 1, if \ m = 1 \end{cases} \qquad (4)$$

Equation 4 was used to create different numbers for the decision-making parameters. The output of the model was then calculated for various numbers within the intervals. This technique created different values for the output of the model.

## 5. Results and Discussion

Multiple scenarios and experiments were designed to evaluate the proposed methods and strategies. The results are presented in this section. In accordance with historical data and experts' opinion, three probable earthquakes were simulated with magnitudes of 6.6, 6.9, and 7.2. Based on the process presented in Figure 2 for different scenarios, the magnitude of the earthquake was calculated at the location of the buildings. Then, based on the vulnerability curve of buildings and the type of materials used, the amount of destruction of each building is determined. Figure 6 shows the vulnerabilities of buildings in these scenarios in the ArcGIS environment.

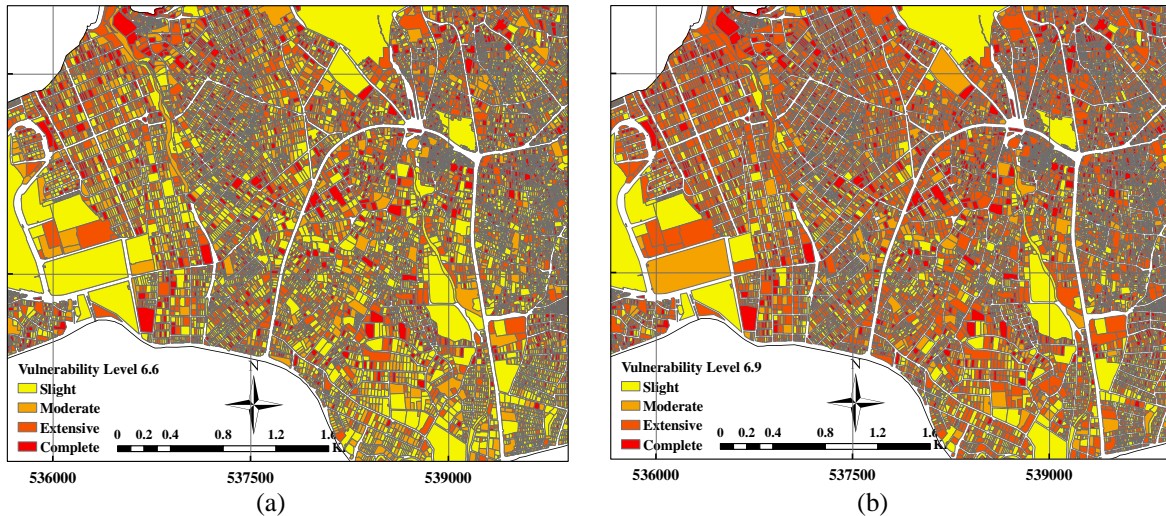

(a)                    (b)

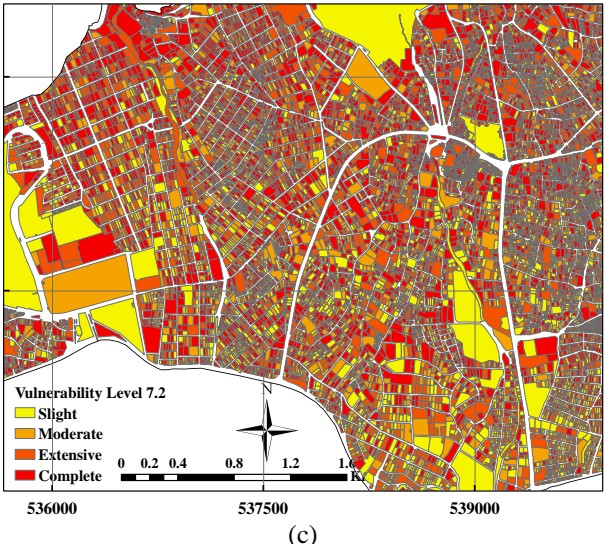

(c)

**Figure 6** Vulnerability maps for District 1, based on earthquakes with magnitudes of a) 6.6, b) 6.9, and c) 7.2 on the Richter scale.

The results of estimating the vulnerability of District 1 for scenario 6.6 show the complete destruction of 18% of the buildings (7,063 buildings of all buildings) in the study area, which is mostly located in the central and northeast part of the region. For this scenario, 27% of the extensive destruction is observed in the buildings, so that these buildings are uninhabitable and there is a possibility of the vulnerability of people in this category of buildings. In the 6.9 magnitude scenario, 29% of complete destruction and 31% of extensive destruction is observed in buildings. It is obvious that with the increase of earthquake intensity, the amount of destruction of buildings increases. Scenario 7.2 shows that with this intensity, 53% of the buildings are severely damaged and these buildings will not be usable. Most of the damaged buildings are located in the central part of the region. A similar result has been obtained in research [22], the main reason being the high structural density and population in this part of District 1. In previous researches, Hashemi and Alesheikh (2011) estimated the number of complete damaged buildings 4% and 32% damage for District 10 of Tehran based on the 6.4 Richter for the Mosha Fault. Hooshagi and Alesheikh (2018) estimated the damage to buildings for the city of Tehran at 16% complete destruction and 24% extensive destruction based on the 6.6 Richter scenario in Niavaran Fault. The degree of degradation of 18%, 29% and 53% according to scenarios 6.6, 6.9 and 7.2 in this study is almost similar to previous researches.

After calculating the vulnerability of buildings and based on the formulas expressed in Figure 2, the numbers of injured and deceased people can be calculated using the JICA model. The numbers of injured and deceased people in scenarios with 6.6, 6.9, and 7.2 magnitude earthquakes are listed in Table 2.

**Table 2** Results of earthquake simulations

| Severity level | Numbers of affected individuals | | |
|---|---|---|---|
| | 6.6 Richter | 6.9 Richter | 7.2 Richter |
| Uninjured | 374,295 | 270,455 | 182,340 |
| Injured | 28,856 | 73,195 | 111,463 |
| Deceased | 30,349 | 89,850 | 139,697 |

According to Table 2, as the magnitude of the earthquake increases, the number of people dead and injured increases, so that in the 7.2 magnitude earthquake, 58% of the people were directly involved. Based on the JICA model, Mansouri et al. (2008) estimated the number of deaths and injuries related to the 6.7 magnitude Riches earthquake at 7% and 4%, respectively. The percentage of people who died and were injured in their research is similar to the 6.6 magnitude earthquake scenario in our research. The computational scale of the JICA model uses

urban blocks. Therefore, the numbers of deceased and injured individuals in each urban block were calculated. The locations of injured individuals were presumed to be in the centers of the respective blocks.

The environmental simulation and proposed method were implemented in AnyLogic software. To simplify the environment and reduce the calculation volume, each agent was regarded as a group in the real world. Figure 7 shows the simulated environment.

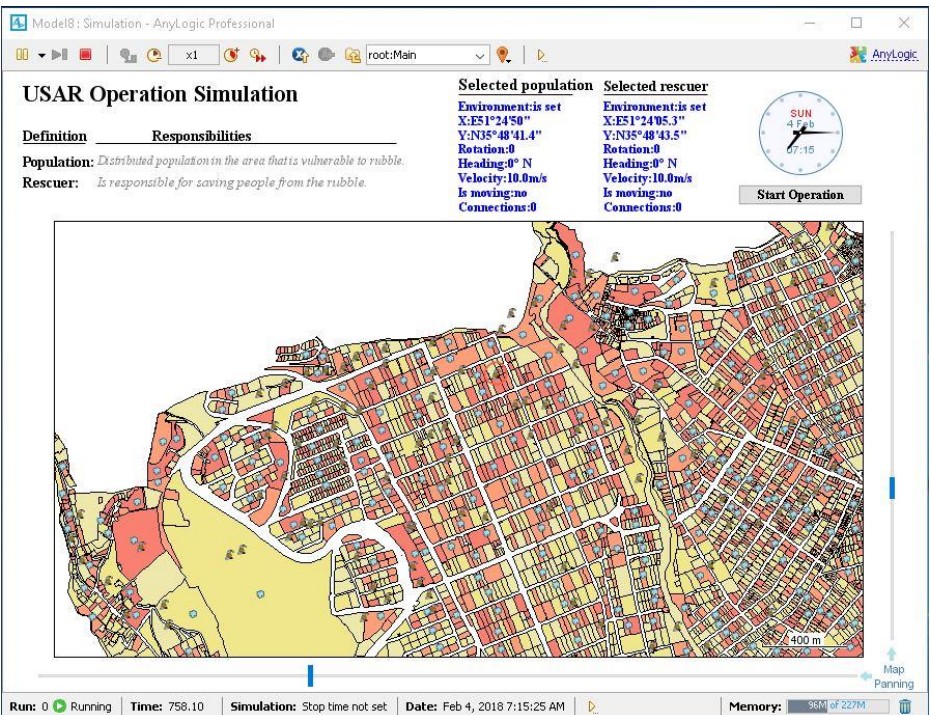

**Figure 7** Overview of the USAR simulator.

Table 3 shows the durations of USAR operations as estimated using scalability analysis with the proposed method. In creating this table, an uncertainty of 30% was considered. For this purpose, the range of task characteristics used the intervals [X, X + 30%X] and [X−30%X, X]. At each stage, a given agent participated in the auction. For that agent's decision-making parameters, the numbers were randomly converted into an interval. The average range of agent tasks and decision-making was used for implementation of the CNP, rather than interval values.

**Table 3** Comparison of operation duration in hours between the proposed method and the CNP (based on 30% uncertainty)

| No. of agents | 1000 | | | 1500 | | | 2000 | | |
|---|---|---|---|---|---|---|---|---|---|
| Simulated earthquake magnitude | 6.6 R | 6.9 R | 7.2 R | 6.6 R | 6.9 R | 7.2 R | 6.6 R | 6.9 R | 7.2 R |
| No. of tasks | 28,856 | 73,195 | 111,463 | 28,856 | 73,195 | 111,463 | 28,856 | 73,195 | 111,463 |
| CNP | 53.16 | 169.03 | 282.76 | 32.83 | 94.24 | 174.19 | 22.6 | 68.95 | 127.47 |
| Strategy 1 | 45.37 | 142.47 | 241.81 | 25.22 | 74.91 | 135.75 | 19.643 | 59.36 | 108.56 |
| Strategy 2 | 44.87 | 137.30 | 234.92 | 26.02 | 76.41 | 138.52 | 19.097 | 58.21 | 105.58 |
| Strategy 3 | 43.75 | 133.76 | 230.12 | 25.75 | 74.33 | 132.75 | 18.332 | 56.33 | 101.77 |
| Strategy 4 | 41.63 | 130.41 | 222.18 | 23.89 | 71.14 | 127.87 | 17.013 | 53.91 | 97.73 |

The operational time decreased when the number of agents in rescue operations increased with the number of tasks remaining fixed. The reduction rate ranged from 54% to 60% when the number of agents was doubled. The duration of a USAR operation increased when the number of tasks increased for a given number of agents.

Therefore, the duration of the rescue operation was related to the number of rescue agents and the number of available tasks in a scenario. There was an inverse relationship between the duration of the USAR operation and the number of rescue agents, and a direct relationship between the duration of the operation and the number of tasks.

The inclusion of uncertainty in any allocation strategy provided better results, compared with the CNP method. Using the proposed strategies, the smallest improvement in results with uncertainty was 2.9 h (13%) for a scenario with 2000 agents and 28,856 tasks (6.6 magnitude earthquake). The maximum improvement was 60.6 h (21%) hours for 1000 agents and 111,463 tasks. In a previous study, task allocation progress was 18% when uncertainty was applied in a laboratory environment regardless of spatial strategies [12]. Previous research has also shown that that taking uncertainty in task allocation into account in District 3 of Tehran improved the duration of the rescue operations by 20%, and decreased the number of fatalities by 15% [1]. Therefore, the proposed approach in this study showed a better performance than the traditional CNP methods.

Among the task allocation strategies in this study, strategy 1 produced the worst response. At each scale for the discussed scenarios, strategy 1 resulted in USAR operations with the longest durations, compared with other strategies. Strategies 1 and 2 provided similar results at different scales, although strategy 2 achieved better results. Strategy 4, which involved spatial information in task allocation, produced better results at all scales including improvements of 21%, 24%, and 23% with 1000 agents for a 6.6 magnitude earthquake, 1500 agents for a 6.9 magnitude earthquake, and 2000 agents for a 7.2 magnitude earthquake, respectively, compared with the CNP. The average improvement for strategy 4 was 26.6 h in rescue operations. The use of strategies 3 and 4 is more suitable in a larger environment with high numbers of both injured people and rescue agents, because controlling agent distribution with respect to expansion of the environment and the uncertain environmental conditions can be effective in future task allocations. In a real-world crisis-ridden environment, the overall environment is damaged and the injured people are well distributed. Therefore, the spatial distribution of agents is an important parameter to control in USAR operations.

The simulation results in terms of deceased people for 1000, 1500, and 2000 agents with different numbers of tasks are shown in Figure 8. In these figures, for each of the four priority parameters and decision parameters associated with agents, a 30% uncertainty level was considered.

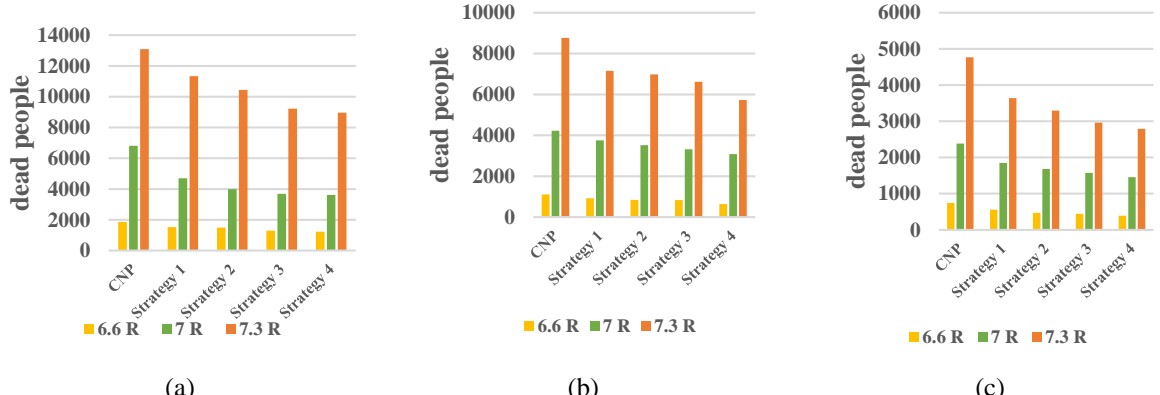

(a)       (b)       (c)

**Figure 8** Numbers of deceased people with a) 1000, b) 1500, and c) 2000 rescue agents.

Figure 8 illustrates the numbers of deceased people in the rescue process with different numbers of agents and tasks. Based on Figure 8, an increased number of tasks led to an increased number of deceased people, but an increased number of rescue agents led to a decreased number of deceased people. Regarding the numbers of deceased people at all three scales, the CNP method produced the worst response. An average of 7253 people were deceased in the CNP model with 1000 agents. Conversely, 5853 people were deceased in the model

employing strategy 1 with 1000 agents. Overall, when all strategies were considered, strategies 4 and 1 resulted in the best and worst responses, respectively. As illustrated in Figure 8, the numbers of deceased people were approximately equivalent in strategies 1 and 2.

Figure 9 illustrates the simulation results for the incorrect allocation of 1000, 1500, and 2000 agents with several different tasks.

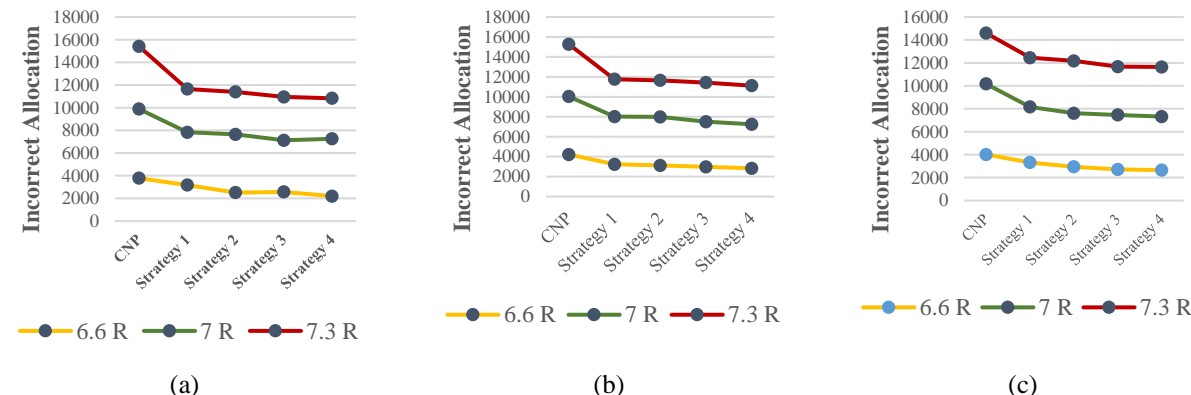

| (a) | (b) | (c) |

**Figure 9** Numbers of incorrect allocations with a) 1000, b) 1500, and c) 2000 rescue agents.

The overall trend in each chart was approximately similar if all charts were considered simultaneously. Any incorrect allocation was unrelated to the number of rescue agents, because there were no changes when the number of agents was increased. The number of incorrect allocations changed with the number of tasks, such that it increased with an increasing number of tasks. This increase is evident in all panels in Figure 9. Incorrect allocations usually occurred at a nearly fixed rate.

Based on the results, the traditional CNP model produced the worst response. The total incorrect allocations in the CNP model with 1000 agents and 28,856 tasks, 1500 agents and 73,195 tasks, and 2000 agents and 111,463 tasks were 3780, 10,027, and 14,604 tasks, respectively. The numbers of incorrect allocations assigned by strategy 1 were 3174, 8014, and 12,455 tasks, respectively. Furthermore, the evaluation criteria showed the advantages of including uncertainty in task allocation. Therefore, the proposed approaches for all three evaluation parameters resulted in better performance, compared with the traditional CNP method. The results indicate that the reallocation of tasks through the proposed approaches and strategies offered a better response, based on the scale of the event, because their differences from the CNP model increased at a larger scale.

The results of interval uncertainty analysis were achieved with 1000 randomized runs of each scenario (Figure 10).

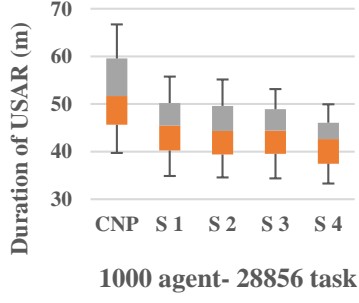

**1000 agent- 28856 task**

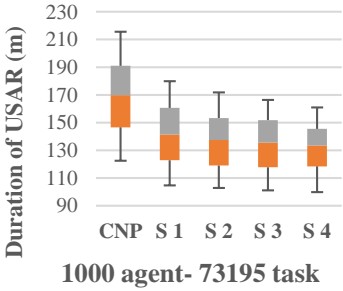

**1000 agent- 73195 task**

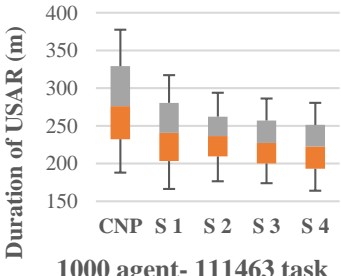

**1000 agent- 111463 task**

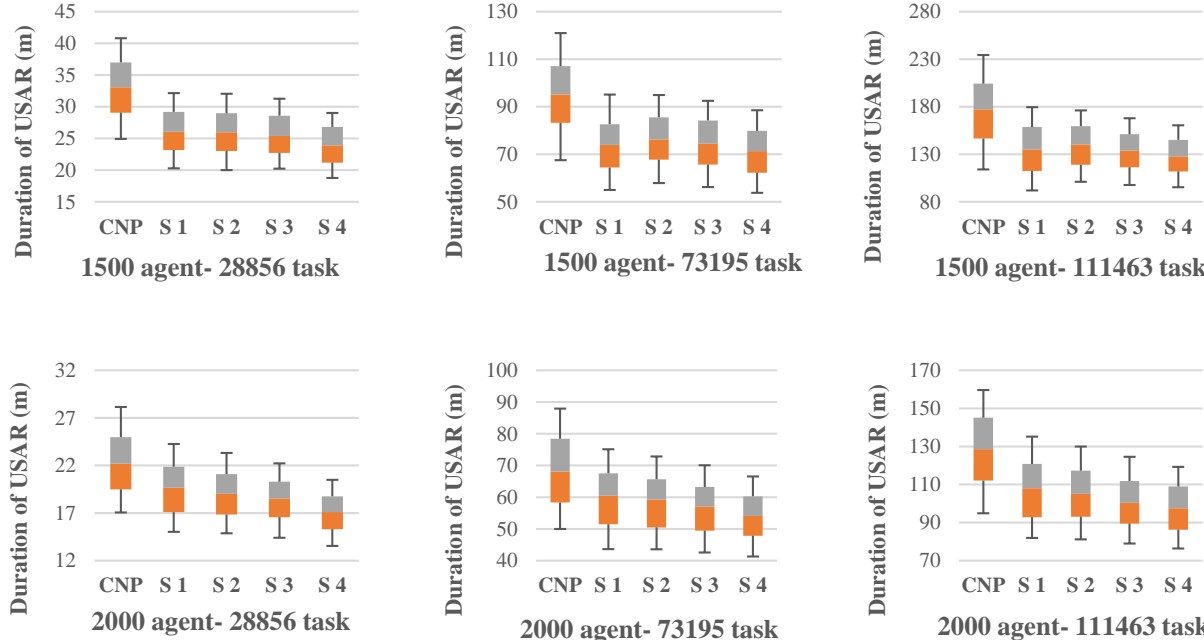

**Figure 10** Uncertainty analysis of the proposed method for USAR operations, for nine simulated scenarios

As shown in Figure 10, there is a direct relationship between interval length and operational time. According to Equation 3, assigning fewer tasks leads to less operating time and causes less uncertainty in the simulated environment.

As mentioned in section 4.3.3, the rescuers use [X, X + 30%X] and [X − 30%X, X] to determine the intervals. Another analysis was performed for values other than 30% in the estimations. The results are shown in Figure 11. An average event scale (1500 agents and 73,195 tasks) was used and different levels of uncertainty (uncertainty between 5% and 55% at five-unit intervals) were randomly generated, investigated, and evaluated. This realistic test aimed to assess the proposed scenarios for each uncertainty value.

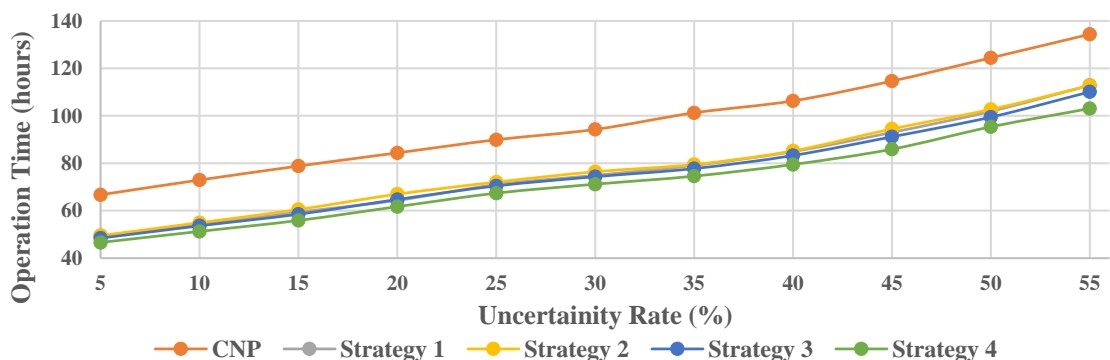

**Figure 11** Uncertainty analysis when different values were used in determining intervals

Figure 11 indicates a relationship between increased in uncertainty (from 5% to 55%) and an increased rescue time. The increases differed among strategies. The increase was 67.7 h for the CNP (from 66.8 h to 134.4 h), whereas increases of 63.4, 63.2, 61.7, and 56.5 h were obtained for strategies 1–4, respectively. Based on the evaluation results, the proposed methods are more efficient and present better responses in the presence of various uncertainties. Therefore, increased in uncertainty leads to a delay in USAR operations and possible task elimination. Accordingly, delaying rescue operations or removing tasks from the rescue list will increase USAR time.

## 6. Conclusion

Providing a suitable method for assigning tasks under uncertain conditions is important, according to the results of simulated USAR operations. This study presented a task allocation approach that aimed to better assign initial tasks, thus ensuring better conditions for potential reallocations of tasks and wasting the least time possible for rescue teams if problems were encountered during the initial allocations or a new task emerges. Some of the characteristics and advantages of the study include the focus on the necessity of task reallocation in disaster environments, the provision of an innovative approach for managing uncertainties that cause non-performance of tasks in the CNP method (the most widely used task allocation method in multi-agent systems), and the definition of spatial strategies for better task reallocation. The proposed approach can be used in combination with a wide range of algorithms for assigning tasks in accordance with the structure of the system.

The results obtained from simulations with the proposed approach revealed that the duration of rescue operations when the proposed strategies were implemented was always shorter than the time required using the CNP method. The worst improvement was identified for 2000 agents with 28,856 tasks (13%) and the best for 1000 agents with 111,463 tasks (21%). Furthermore, the results at different scales showed that the application of uncertainty in task allocation could improve the duration of USAR operations. There is a relationship between increased in uncertainty and increased rescue operation duration. Furthermore, the results revealed a significant decrease in the numbers of deceased people and wrong allocations due to uncertainties, which demonstrated the importance of uncertainty inclusion in task allocation. The implemented method can be used for cooperation among agents. In an earthquake-stricken environment, rescuers can use assistant agents (devices such as mobile phones and tablets) to implement this methodology.

However, regarding comparisons of the proposed strategies, it is insufficient to consider only uncertainty in initial decision-making concerning task allocation because the working environment is quite dynamic and the assigned tasks may encounter various problems. An effective assignment approach should consider both uncertainties in decision-making and strategies for reallocation to waste the least time during system disruptions. This optimizes planning to achieve better implementation time and allows fault tolerance. The strategies for applying uncertainty during the implementation of task allocation improve the efficiency, performance, and stability of agent-based cooperation. Task allocation strategies lead to flexibility in decision-making and decrease the system's overall costs. Furthermore, spatial task allocation strategies can propose a specific arrangement of the rescue team within an environment to prevent time-wasting in the event of environmental uncertainties or task reallocation.

One of the limitations of agent-based simulation is the difficulty of implementation and time-consuming processes that require the availability of powerful processors. Although the proposed method is simple in terms of interval uncertainty and does not perform complex calculations, it is time-consuming due to a large number of calculations to apply spatial strategies for each task and each agent. These processes require the availability of powerful processors. Another limitation of the proposed method is the assumption of communication between agents and the central agent. The proposed method is developed for assistant agents in which groups' information (as agents) is transmitted between groups through tools such as mobile phones. Although the volume of message transfer in this method is less than the traditional CNP method, in severe earthquakes that damage the internet infrastructure, the task allocation method still has problems. This limitation exists in all communication methods that consider the whole groups.

In this study, a simplified environment was considered, for example, it was assumed that the roads were not damaged. It is suggested that environment simulators such as Hazus or CIPCast-ES be used as a deterministic approach to simulate the damaged environment and the vulnerability of roads and. Future studies could also

present the output of this study in the form of a spatial decision support system (SDSS) or a webGIS. Additional research is recommended to provide new strategies and combine the proposed task allocation strategies of the present study with a coalition-forming method to select an appropriate coordinating agent in our proposed approach.

## 7. Acknowledgments

This research was conducted by the Basic Research Project of the Korea Institute of Geoscience and Mineral Resources (KIGAM) funded by the Ministry of Science and ICT.

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
