# Peer review of "USAR simulation system: spatial strategies for agent task allocation under uncertain conditions"

_Natural Hazards and Earth System Sciences, 2020_

## Referee Comment (RC1) · Anonymous Referee #1 · 20 Nov 2020

Dear Editor Thank you so much for considering me to review the mentioned article

This paper proposes an agent-based simulation model to investigate uncertainty in tasks allocations in urban search and rescue (USAR) operation. The paper has an interesting and relevant topic to this journal. Although, the paper provides a clear image of the performed research and has a sound experimental work, my concern lays in its profession English writing even by a large effort that went into this study. It is believed by this reviewer that the manuscript deserves publication once the following minor comments are observed.

1- The innovation of the article should be explicitly stated in the abstract, as well as in

the introduction, the volume of the article can be reduced by deleting the general and repetitive sentences.

2- In the abstract, the numerical results should be expressed as percentages to make them more understandable.

3- Authors are requested to elaborate on how their proposed method can be used in the real disaster environment? Do rescuers need to use mobile phones and tablets as an assistant agent?

4- In the text, either uses the word reallocation or replanning.

5- State the references used for the following sentence or argue on its reasons.

"Methods such as simulated annealing (SA) and the ant colony optimization algorithm cannot find a global optimization of the problem and provide local solutions instead."

6- In the implemented method, express what happened if a task is not executed, are the new values definitely considered or re-entered into the cycle with uncertainty?

7- State the units used in Equation 1 for distance, etc.

8- More explanation of Figure 3 is needed.

9- How did you create the real numbers in step 5 of the proposed method? Discuss more.

10- Correct Equations numbering.! Equation 1 exists in two parts.

Please also note the supplement to this comment:
https://nhess.copernicus.org/preprints/nhess-2020-277/nhess-2020-277-RC1-supplement.pdf

---

## Referee Comment (RC2) · Anonymous Referee #2 · 4 Dec 2020

This paper provides an interesting perspective in the use of multi-agent simulation for the rescue in areas submitted to an earthquake. I think that paper could be shortened and the text should be precised. In its present form, I find difficult to understand what is multi-agent simulation reading this paper. I feel confuse about the scope of the first part (part 2) of the paper. Is it for explaining what is multi agent simulation? If it is the case, I (as reader) am not able to understand what is multi agent simulation. Or Is it dedicated to specialists of multi-agent simulation? if it is the case, the part concerning the general use of multi agent simulation has to be removed and the bibliography of the 2.3 should be developed (from line 156 to 176). Part 3 (case study): What data are available for this case study? Line 194 "The proposed methodology is a general approach to

various phenomena." This sentence is empty. Please remove or be clearer. Line 195: What are the characteristics of the environment which are known? What are the unknowns? Line 196: what controls the uncertainty for a person to be trapped and injured? How is it decided? Is it spatially controlled? Line 205: What can be the disruptions? Are there statistically defined? Line 210: I do not understand how injured agents can communicate with other agents. It is possible that injured agents blocked under the destroyed buildings are conscious and able to make some noise but this is rare. Line 260: "these relationships are based on expert opinion". Could you add a reference?

In its present form, I do not recommend this paper for publication. Authors have to defined the audience for who they write, they have to be more precise in their description, and they have to shorten the paper

––––––––––––––––––––––––––––––––––––––

---

## Author Comment (AC1) · 28 Jan 2021

This paper provides an interesting perspective in the use of multi-agent simulation for the rescue in areas submitted to an earthquake. I think that the paper could be shortened and the text should be precised. In its present form, I find difficult to understand what is multi-agent simulation reading this paper.

Response: We are grateful for the opportunity to explain our manuscript. We thank you for providing highly constructive and insightful comments to improve our manuscript. Based on the reviewer's comments, the editor's opinions, and deep thought in the article, it was decided to change the writing structure of the article. Numerous details

were included in the text that made it difficult to understand. For correction, we first explain the whole research in the manuscript and introduce its three phases. We also state that the first and second phases are stated in our previous articles, and in this article, only the intended innovation is described. Interested parties are referred to our previous articles to understand the concepts. As mentioned, this article is the final part of a research project in Iran. This research has been done in three phases: In the first phase, uncertainty in the task allocation among agents is considered and task allocation is done only by considering the proximity (spatial distance) to the tasks. The developed method was evaluated in a square-shaped random environment and no sensitivity analysis was performed. The results were presented in an article entitled "Agent-based task allocation under uncertainties in disaster environments: An approach to interval uncertainty". In the second phase, the feasibility of the developed method in the real environment is investigated. In this paper, the aim was to simulate the operational environment of the crisis and to examine the developed method in the real environment. In the simulated system, the 6.8 Richter earthquake damage was calculated for District 3 of Tehran, and rescue operations were modeled. The results were presented in an article entitled "Developing an agent-based simulation system for post-earthquake operations in uncertainty conditions: a proposed method for collaboration among agents". In the third article (submitted article to NHESS), spatial strategies are included in tasks allocation among agents and simulated with the real environment data. Tasks allocation in crisis environments has spatial nature and the location of the injured and rescue agents play a key role in the assignment of tasks. Therefore, while considering the uncertainty in tasks allocation among agents (the subject of the first article) and simulating the method in the environment with real data (the subject of the second article), different task allocation strategies are examined, and the accuracy of the previous methods increased significantly. The study area in this article is also changed to District 1 of Tehran to evaluate the method capability in different study areas. Our submitted article is a combination of the previous two articles with spatial strategies' innovation in it. In the present article, although there are concepts

and practical points of the previous two articles, the main innovation of the research is in terms of spatial strategies and also evaluations are based only on spatial strategies. By applying these corrections, the volume of the article will be reduced and the readers will study the article with a certain intellectual background, and therefore it will be easy to understand the article. The sections that have been fully explained in our previous articles have been summarized and readers have been referred to those articles. Also, new sections and innovations of the article were explained in more detail. The changes are marked by the track and change tool in Word.

In Section 1. Introduction: "The present article is the final part of a research project in Iran. This research project was carried out over three phases. In the first phase, uncertainty in task allocation among agents was considered and task allocation was performed only by considering the proximity (spatial distance) to the tasks. The developed method was evaluated in a square-shaped random environment without a sensitivity analysis [12]. In the second phase, the feasibility of the developed method was investigated in a simulated environment using real regional data. In this phase, the operational environment of a crisis was simulated and the developed method was examined in a real environment. In the simulated system, damage for a 6.8 magnitude earthquake damage was calculated for District 3 of Tehran, and rescue operations were modeled [1]. In the third phase using the concepts of previous articles [1, 12], spatial strategies were included in task allocation among agents and simulated with real-environment data. The present paper is the output of the third phase of the research project, which aimed to improve task allocation in crisis-ridden conditions for agent-based groups by considering proper strategies to manage uncertainties. This paper first develops an agent-based simulation system for USAR operations, then applies uncertainties in agent decision-making by improving an interval VIKOR method to perform task allocation, and defines strategies for conditions under which the initial assignment has encountered a problem and requires reallocation (i.e., managing availability uncertainty during implementation). The main innovation of the study is the establishment of an approach to improve conditions during reallocations or future allocations when initial allocations encounter problems due either to availability uncertainties or the addition of a new task. In general, strategies are selected in such a manner that the final cost of the system will not increase abnormally if the initial allocations encounter problems."

I feel confuse about the scope of the first part (part 2) of the paper. Is it for explaining what is multi agent simulation? If it is the case, I (as reader) am not able to understand what is multi agent simulation. Or Is it dedicated to specialists of multi-agent simulation? if it is the case, the part concerning the general use of multi agent simulation has to be removed and the bibliography of the 2.3 should be developed (from line 156 to 176).

Response: This insightful comment is highly appreciated. In this section, it is assumed that users have prior knowledge of multi-agent systems (MAS) and only the various applications of MAS were mentioned. Based on the comments of the reviewer and the fact that readers of the NHESS article may not know MAS, general applications were omitted and multi-agent systems were briefly described and readers were referred to other articles to study the general use of MAS. This section was edited as follows: Section 2.1. Agent-based USAR simulation: "An agent-based model is a class of computational models for simulating the actions and interactions of autonomous agents. Agent-based simulations have been used in various investigations including crisis/disaster management [1, 16], emergency supply chains [17], tsunamis [18], and collective behavior [19]. These simulations can be effective in both planning and policymaking [20]. Simulation of the operating system involves a simplified real environment, which is used to model a wide range of agents in complex systems. Various researchers have modeled a portion of the behavior of agents in simulated environments [16, 18, 21] and demonstrated collaboration among agents. However, agent cooperation in catastrophic environments has been less extensively studied, such that uncertainty in collaboration among agents has generally not been considered. In previous studies, a geospatial information system platform was used when preparing the environment and creating

a simulation base map [19]. Spatial analysis and tools related are used in most research endeavors in USAR operations after an earthquake." Also, the bibliography of the 2.3 Reallocation and reassigning methods developed as follows: "Distinct algorithms have been proposed for scheduling and task reallocation in accordance with the tasks and available conditions within an environment [34]. Some reallocation methods (e.g., data envelopment analysis [35]) and exact algorithms (e.g., a branch-and-bound algorithm with column generation) resolve problems on a smaller scale (e.g., 10 jobs and three vehicles). In such methods, the process is time-consuming and slow for resolving large-scale problems [13]. Therefore, they are not suitable for the allocation of tasks that should be performed dynamically and instantaneously in large-scale problems. In some research, such as the investigation of gate reassignment problems, initial assignment tables have been created using heuristic methods in such a manner that a succession delay is minimized [36]. The incidence of adverse events may disrupt the original table. Notably, this method is not suitable for a large number of tasks. Some other task allocation methods are interdependent with the plan's ongoing tasks, such as in construction operations [14]. In those mathematical calculations, when a task fails, all other tasks that were based on its correct implementation must be replanned. An appropriate reallocation method must be applied with respect to the nature and scale of the problem. In USAR, a rescue process generally occurs independently of any other rescue processes, and only a portion of the workflow is ready to be implemented and assigned. Moreover, because of the large number of rescue groups in USAR operations, as well as the available uncertainties and dynamic nature of multi-agent systems in disaster environments, the concept of general planning is uncommon and appropriate plans should be produced both locally and cross-sectionally. Most available methods to resolve the problem of assigning tasks cannot be developed for uncertain conditions and restrictions such as in critical rescue environments (e.g., USAR after earthquakes). With respect to USAR operations, task allocation methods must include different strategies for all conditions and be dynamically generated in a real-time environment. In contrast to previous studies, we define an approach based

on spatial strategies, such that the results of the initial task allocation are used for future task allocations and are appropriate in the rescue environment. Time limitations constitute another issue in the reallocation and reassignment of rescue teams. Therefore, the present study aims to expand the CNP method for rapid problem resolution."

Part 3 (case study): What data are available for this case study?

Response: We generally agree with the reviewer's point to add a data section. In Iran, integrated data from regions is rarely and hardly found. The following paragraph was added to the text to specify the data used. Section 3. Case study and data: "The basic data used in environment simulation were block maps, population, distance from the fault, building material, agent location, year of building construction, and building height."

Line 194 "The proposed methodology is a general approach to various phenomena." This sentence is empty. Please remove or be clearer.

Response: Thank you for your in-depth analysis. The developed method is suitable for cooperation between agents in different phenomena in which agents are in relation to each other. For example, this method is suitable for cooperation between agents in rescue operations during floods, terrorist attacks, and other operations in which several agents must cooperate. Since this case has not been studied in our research, this sentence is removed from the manuscript.

Line 195: What are the characteristics of the environment which are known? What are the unknowns?

Response: Thank you for pointing out this misunderstanding to us. By this, we mean simply to say that there is uncertainty in the environment and that environmental information is not entirely clear. For example, the travel time from point A to B is uncertain or, for example, the exact number of casualties in urban blocks is not certain. Items in which uncertainty was considered to include the number of injuries, the severity of the

victims' injuries, duration of the operation, infrastructure priorities, agent energy, route status, task runtime by an agent, and risk level for the agent. Therefore, the sentence is edited as follows. Section the scenario of proposed agent-based USAR simulation: "We assume the presence of a disaster environment in which events are uncertain." Also, "Given the results of previous studies [12, 33, 39, 40] and in accordance with expert opinion on USAR operations, the uncertainties include the number of injuries, severity of the victims' injuries, duration of the operation, infrastructure priorities, agent energy, route status, task runtime by an agent, and risk level for each agent. These are important uncertainties in task allocation. All parameters are specified as intervals during the task allocation process."

Line 196: what controls the uncertainty for a person to be trapped and injured? How is it decided? Is it spatially controlled?

Response: Thank you for your in-depth analysis. There is a population distribution map of the area as shown below.

(Figure 1)

Based on this map and the JICA model, the number of injured people in each urban block is determined. The JICA methodology has four major stages: namely, seismic hazard assumption as an input, building inventory development, building, and human vulnerability function developments and implementations, and, finally, the production of results in a GIS [17]. The inputs of the model are building material, building height, a building's year of construction, distance from the fault, and parcel maps and fragility curves. To calculate the number of injured people, building population and the following Equation is used [17]:

Figure 2- Equation (1)

This number does not exist exactly in the urban block. For example, part of this population goes to work during the day, some are out of the house at night, so the population

in urban blocks is not always a fixed number. Therefore, according to the formulas, there will be uncertainty in the number of injured in each urban block. References to previous articles were given for how to calculate these values. The sentences were edited as follows to clarify the subject.

Section 4.1. The scenario of proposed agent-based USAR simulation: "The injured individuals are trapped under rubble and the number of such individuals in each building block is uncertain. Rescuing injured people is the main goal. Saving each person is a task that must be performed through the cooperation of rescue agents. After an earthquake, the numbers of injured and deceased people can be estimated by using different formulas by determining the magnitude and location of the earthquake, as well as the urban context data of the buildings [38]. Furthermore, the possible locations of injured individuals can be predicted using building damage assessment models. Therefore, the simulation inputs are the injured individuals' locations and their characteristics, which are available with some uncertainty."

Section 4.2. USAR simulation: "To simulate an earthquake-damaged environment, an earthquake risk assessment model was developed based upon the Japan International Cooperative Agency (JICA) model. The JICA model is the output of cooperation between the Center for Earthquake and Environmental Studies of Tehran and the JICA. The results of this project and its implementation have been presented previously [41] and used in various studies [1, 42]. This model can calculate the buildings' level of destruction and the number of injured people based on the earthquake intensity, earthquake location, building vulnerability, and the population in these buildings."

Line 205: What can be the disruptions? Are there statistically defined?

Response: We appreciate the reviewer's question. In this article, tasks allocation is considered with uncertainty. So any big difference from the initial interval can be considered as disruption (in the last part of Figure 3,4 [which is edited as follows]). For example, the initial evaluations show that the route is safe, while the agent realizes

when he is in the area that it is practically impossible to move towards the desired route. Or, for example, the initial estimate of the number of injured people in a house is between 3 and 5 people, and the agent goes to the area and sees that the number of injured people is fifteen. Certainly, their equipment will not be enough and they may not be able to work due to limitations. Therefore, he requests the reallocation of the work.

(Figure 3 Task allocation flowchart in the proposed approach by five steps and environmental simulation)

(Figure 4 Task allocation flowchart in the proposed approach, separated into five steps within an environmental simulation)

The disruptions implementation and application of them in formulas is described in Section 4.2.5 Implementation and observation of real values in the environment, which was edited as follows to be understandable. Section 4.3. 5. Implementation and observation of real values in the environment: "During the implementation phase, tasks are implemented by agents in a dynamic environment where there are always uncertainties during task execution. The rescuer observes the difference between predicted values and the actual environment after the work begins. In this study, a random number in the $[X -30\%X, X + 30\%X]$ interval was chosen to model the real environment. In the real world, the difference between the predicted environment (through building vulnerability estimation models) and the real environment will determine the agent's performance. If the agent observes a large difference between the auction information and the real environment, the agent abandons that task. In this instance, the agent updates the task's values and uncertainties and returns the work to the central agent. The new uncertainty interval will be 80% smaller than the original interval. There are various conditions under which agents will reallocate a task if the environment differs from the expected scenario. For example, the agent can abandon the task if three of eight decision-making parameters are out of range by 5%. Otherwise, the agent finishes the rescue work by accepting the new conditions. The central agent assigns

newly added tasks within the reallocation framework. When a new task is assigned, the task allocation is combined with that of both new and incomplete tasks."

Line 210: I do not understand how injured agents can communicate with other agents. It is possible that injured agents blocked under the destroyed buildings are conscious and able to make some noise but this is rare.

Response: Of course, the injured person cannot have any interaction. In the proposed model, there is an injured agent without any communication in the environment and only its vital signs are changing with a constant trend. Other agents interact with each other. To make the text clearer, emphasize that the injured person has no activity in the environment. The following sentence was included in the text. Section 4.1. The scenario of proposed agent-based USAR simulation: "This agent exists in the environment and has a critical condition that changes continuously. This agent has no activity or communication with other agents."

Line 260: "these relationships are based on expert opinion". Could you add a reference?

Response: Unfortunately, limited research has been done in this field, and only equivalents are mentioned in Dr. Alireza Vafaeinejad's doctoral dissertation entitled "Spatio-Temporal Modeling and Planning of Working Groups in an Activity-Based GIS (case study: rescue groups)" and the book of rescue operations entitled "Team Forming and Teamwork in Rescue Operations (with emphasis on urban search and rescue team)" in the Persian language. These equations have been after various analyzes with rescue experts. We have also used these formulas in our previous articles [1, 2] , but have not presented them in the text.

In its present form, I do not recommend this paper for publication. Authors have to defined the audience for who they write, they have to be more precise in their description, and they have to shorten the paper.

Response: We believe that our manuscript is substantially improved and has no similarity to our previous articles. We would be glad to respond to any further questions and comments that you may have. Yours Sincerely

References:

1. Hooshangi, N. and A.A. Alesheikh, Agent-based task allocation under uncertainties in disaster environments: An approach to interval uncertainty. International Journal of Disaster Risk Reduction, 2017. 24: p. 160-171.

2. Hooshangi, N. and A.A. Alesheikh, Developing an Agent-Based Simulation System for Post-Earthquake Operations in Uncertainty Conditions: A Proposed Method for Collaboration among Agents. ISPRS International Journal of Geo-Information, 2018. 7(1): p. 27.

3. Wang, Y., K.L. Luangkesorn, and L. Shuman, Modeling emergency medical response to a mass casualty incident using agent based simulation. Socio-Economic Planning Sciences, 2012. 46(4): p. 281-290.

4. Ben Othman, S., et al., An agent-based Decision Support System for resources' scheduling in Emergency Supply Chains. Control Engineering Practice, 2017. 59: p. 27-43.

5. Erick, M., et al., Agent-based Simulation of the 2011 Great East Japan Earthquake/Tsunami Evacuation: An Integrated Model of Tsunami Inundation and Evacuation. Journal of Natural Disaster Science, 2012. 34(1): p. 41-57.

6. Welch, M.C., P.W. Kwan, and A.S.M. Sajeev, Applying GIS and high performance agent-based simulation for managing an Old World Screwworm fly invasion of Australia. Acta Tropica, 2014. 138, Supplement: p. S82-S93.

7. Fecht, D., L. Beale, and D. Briggs, A GIS-based urban simulation model for environmental health analysis. Environmental Modelling & Software, 2014. 58: p. 1-11.

8. Matarić, M.J., G.S. Sukhatme, and E.H. Østergaard, Multi-robot task allocation in uncertain environments. Autonomous Robots, 2003. 14(2-3): p. 255-263.

9. Gokilavani, M., S. Selvi, and C. Udhayakumar, A survey on resource allocation and task scheduling algorithms in cloud environment. ISO 9001: 2008 Certified International Journal of Engineering and Innovative Technology (IJEIT), 2013. 3(4).

10. Barnum, D.T. and J.M. Gleason, DEA efficiency analysis involving multiple production processes. Applied Economics Letters, 2010. 17(7): p. 627-632.

11. Cai, B., et al., Rescheduling policies for large-scale task allocation of autonomous straddle carriers under uncertainty at automated container terminals. Robotics and Autonomous Systems, 2014. 62(4): p. 506-514.

12. Cheng, Y., A knowledge-based airport gate assignment system integrated with mathematical programming. Computers & Industrial Engineering, 1997. 32(4): p. 837-852.

13. Olteanu, A., et al., A dynamic rescheduling algorithm for resource management in large scale dependable distributed systems. Computers & Mathematics with Applications, 2012. 63(9): p. 1409-1423.

14. He, Y.H., et al. Research of Allocation for Uncertain Task Based on Genetic Algorithm. in Advanced Materials Research. 2014. Trans Tech Publ.

15. Sang, T.X., Multi-criteria decision making and task allocation in multi-agent based rescue simulation. Japan Graduate School of Science and Engineering, Saga University, Japan, 2013.

16. Chen, A.Y., et al., Supporting Urban Search and Rescue with digital assessments of structures and requests of response resources. Advanced Engineering Informatics, 2012. 26(4): p. 833-845.

17. Mansouri, B., K. Hosseini, and R. Nourjou. Seismic human loss estimation in

Tehran using GIS. in 14th World Conference on Earthquake Engineering, Beijing. 2008.

18. Kang, H.-s. and Y.-t. Kim, The physical vulnerability of different types of building structure to debris flow events. Natural Hazards, 2016. 80(3): p. 1475-1493.

19. Mansouri, B., K. A Hosseini, and R. Nourjou, SEISMIC HUMAN LOSS ESTIMATION IN TEHRAN USING GIS. 2008.

20. Vafaeinezhad, A., et al., Using GIS to Develop an Efficient Spatio-temporal Task Allocation Algorithm to Human Groups in an Entirely Dynamic Environment Case Study: Earthquake Rescue Teams. 2009. 66-78.

[Figure]

[Figure]

**Fig. 1.**

$$\begin{bmatrix} Uninjured \\ Injured \\ Dead \end{bmatrix} = \left(\frac{Population}{Buildings}\right) \begin{bmatrix} -0.073 & 1.040 & 0.650 \\ 0.071 & 0.047 & 0.062 \\ 1.001 & -0.087 & 0.289 \end{bmatrix} \begin{bmatrix} Slight \\ Moderate \\ Extensive + Complete \end{bmatrix}$$

**Fig. 2.**

**Inputs**

| Actual data in environment | Available Tasks with interval uncertainty | Location of rescuers (free workers) |

**Method of applying uncertainty**

**Uncertainty criteria**

**Uncertainties involved in Decision making**
- Agent energy
- Route status
- Task runtime by an agent
- Risk level for agent

**Uncertainties involved in Prioritization**
- Number of injuries
- Severity of injuries
- Duration of operation
- Infrastructure priorities

Apply in the prioritization

Apply in decision making

**Applying in simulation environment**

**Implementation in environment**

- Start implementation process by moving up to tasks
- View the environment and the actual values of uncertainties

- Evaluating the environment and decision making parameters

- A problem in allocation?
  - NO → Do task and release the rescuer
  - YES → Update features of work → Resend it to central agent

**Task allocation method**

Sufficient number of agents? — NO → Wating

YES

**Ordering existing tasks**
- Create decision matrix for four interval data
- Applying the interval VIKOR and sorting the tasks

**Finding coordinating agent**
- Determining n closest agent for high priority tasks
- Choose the closest agent as the coordinator

**Holding an auction**
- Sending the Tasks to coordinator agent
- Calculate decision parameters by itself / Declaring tasks to rescue agents
- Receive agent bids
- Interval Deciding And finding a list of winner agents according to bids
- Announce the ordered list to central agent

**reassigning strategies**
- Applying reallocation strategies
- Announce results to rescue agents

**Fig. 3.**

[Figure]

**Fig. 4.**

---

## Author Comment (AC2) · 28 Jan 2021

This paper proposes an agent-based simulation model to investigate uncertainty in tasks allocations in urban search and rescue (USAR) operation. The paper has an interesting and relevant topic to this journal. Although, the paper provides a clear image of the performed research and has a sound experimental work, my concern lays in its profession English writing even by a large effort that went into this study. It is believed by this reviewer that the manuscript deserves publication once the following minor comments are observed.

Response: Thank you for your in-depth analysis. Although the manuscript has been

edited by "Academic Proofreading Services Ltd" for any spelling and grammatical errors before submitting the article. After applying the corrections intended by the reviewers, it was sent to "Text check - English Consultants" again for English editing, so two natives reviewed and corrected the article. Certification is sent in attachment.

1- The innovation of the article should be explicitly stated in the abstract, as well as in the introduction, the volume of the article can be reduced by deleting the general and repetitive sentences.

Response: This insightful comment is highly appreciated. In order to express the research innovation, the following sentence added to the abstract and introduction section. The article was reviewed and repetitive and general sentences were removed to reduce the volume of the article. The changes are marked by the track and change tool in Word. In Abstract: "Applying allocation strategies is the main innovation of the method." In Section 1. Introduction: "The main innovation of the study is the establishment of an approach to improve conditions during reallocations or future allocations when initial allocations encounter problems due either to availability uncertainties or the addition of a new task."

2- In the abstract, the numerical results should be expressed as percentages to make them more understandable.

Response: The authors completely agree with the reviewer's comment and have revised the sentences as follows to make the results understandable. In Abstract: "Interval uncertainty analysis and comparison of the proposed strategies showed that increased uncertainty led to increased rescue time for the CNP and strategies 1 to 4. The time increase was less with the uniform distribution strategy (strategy 4) than with the other strategies."

3- Authors are requested to elaborate on how their proposed method can be used in the real disaster environment? Do rescuers need to use mobile phones and tablets as an assistant agent?

Response: The implemented method can be used for cooperation between different agents. In crisis environments, rescue teams use assistant agents. These agents can be as software on a mobile phone or tablet. Thank you for pointing out this concern to us. In Section 6. Conclusion: "The implemented method can be used for cooperation among agents. In an earthquake-stricken environment, rescuers can use assistant agents (devices such as mobile phones and tablets) to implement this methodology."

4- In the text, either uses the word reallocation or replanning.

Response: The term "reallocation" was changed to "replanning".

5- State the references used for the following sentence or argue on its reasons. "Methods such as simulated annealing (SA) and the ant colony optimization algorithm cannot find a global optimization of the problem and provide local solutions instead."

Response: The reference of the stated sentence was added. In Section 2.3. Reallocation and reassigning methods section: "Methods such as simulated annealing (SA) and the ant colony optimization algorithm cannot find a global optimization of the problem and provide local solutions instead [12]."

5- In the implemented method, express what happened if a task is not executed, are the new values definitely considered or re-entered into the cycle with uncertainty?

Response: The following sentences were added to the text to clarify the subject. In Section 4.3.5. Implementation and observation of real values in the environment section: "If the agent observes a large difference between the auction information and the real environment, the agent abandons that task. In this instance, the agent updates the task's values and uncertainties and returns the work to the central agent. The new uncertainty interval will be 80% smaller than the original interval."

6- State the units used in Equation 1 for distance, etc.

Response: The following sentence was added to state the units used in Equation 1. In Section 4.3.3. Holding an auction section: "In Equation 1, the distance (in meters),

severity of the victims' injuries, and task priority are based on values declared by the central agent."

7- More explanation of Figure 3 is needed.

Response: We generally agree with the reviewer's point to add more explanation in Figure 3. The following sentence was added to the text. In Section 4.3.4. Applying allocation strategies section: "In Figure 3, a rescue agent located centrally has a strategic position and will try to maintain this position. Although the total movement may increase, if there are problems in performing other tasks, this agent can help all other groups."

8- How did you create the real numbers in step 5 of the proposed method?

Response: Due to the fact that in the real world it was not possible to evaluate the model, simulated values were used. Random numbers in [X - 30%X, X + 30%X] interval was used to create the values of the simulation environment. In Section 4.3.5. Implementation and observation of real values in the environment section: "In this study, a random number in the [X - 30%X, X + 30%X] interval was chosen to model the real environment."

9- Correct Equations numbering.! Equation 1 exists in two parts.

Response: Thanks to this statement, the equation numbering was corrected.
* * *

---

## Author Response (AR2)

Dear Editor,
Dr. Paolo Tarolli
Natural Hazards and Earth System Sciences
July 31, 2021
Title: "USAR simulation system: presenting spatial strategies in agents' task allocation under uncertainties"

We are grateful for the opportunity to resubmit our manuscript to the NHESS. We thank the reviewers for providing highly constructive and insightful comments to improve our manuscript. We have responded in detail to each comment and applied significant changes to our manuscript, based on the reviewers' suggestions. The major changes made are as follows:

1. The role of spatial data and tools in the literature review and background section was described. In this regard, the vital role of accurate and current spatial was expressed in earthquake-affected environments, such as assessing the extent of damage and casualties, surveying vulnerable areas after an earthquake, examining the existing infrastructure in the region, and finally tremendous aid in humanitarian efforts.

2. The history of earthquakes in the study area was studied and the reason for simulating earthquakes with magnitudes of 6.6, 6.9, and 7.2 was explained. In this regard, the existing articles from the study area (District 1 of Tehran) were referenced.

3. We have revised and rewritten the methodology section. We have improved this section with more information about VIKOR and JICA models. The reason for using the VIKOR method and how to use it were written. Also, to make the steps of preparing vulnerability maps more transparent using the JICA model, a figure and a paragraph were added in the methodology section.

4. In order to improve the results section, the discussion regarding the output of the vulnerability of buildings, the number of injured people, and the discussion regarding the output of the proposed task allocation method were added to the manuscript. At this stage, the output of the present study was compared with the output of previous research in each stage.

5. The main limitations of the proposed method were expressed in the conclusion section and based on the existing limitations; several recommendations were provided for future researches.

The next section contains our point-by-point responses to the reviewers' comments. We believe that our manuscript is substantially improved and is more readable for broader audiences. We look forward to hearing from you. We would be glad to respond to any further questions and comments that you may have.

Yours Sincerely

**Reviewer 3**

My revision follows two other reviews' reports provided by the other two experts, and following that the authors revised this manuscript accordingly.

1- **About the article content, first I have to recommend the authors to improve the aim and the provided novelty of the present manuscript. Moreover, a brief synthesis of the expected improvements provided by the proposed research would be welcomed. In this latter case, just one or two sentences could be sufficient.**

Response:

We are grateful for the opportunity to explain our manuscript. We thank you for providing highly constructive and insightful comments to improve our manuscript. To improve the purpose of the research and the innovation of the research, the following sentences were inserted in the text. Then, the expected improvements were expressed after applying the method and achieving the goal and innovation of the research.

In the introduction section: "The main purpose of the research is to improve task allocation in crisis-ridden conditions for agent-based groups by considering proper strategies to manage uncertainties."

In the introduction section: "The innovation of the study is the establishment of an approach to improve conditions during reallocations or future allocations when initial allocations encounter problems due either to availability uncertainties or the addition of a new task. In general, spatial strategies are selected in such a manner that the final cost of the system will not increase abnormally if the initial allocations encounter problems."

In the introduction section: "By applying spatial strategies in the assignment of tasks, it is expected that the tasks allocation in conditions of uncertainty will be done optimally and USAR operations will be performed more quickly."

2- **About the literature review, the authors introduced the role of spatial data and tools in this research topic, too briefly. By the way, geoinformation plays a primary role in earthquake-induced urban rubble as well as well as in earthquake simulation on urban areas. Therefore, a more in-depth analysis should be provided.**

Response:

We generally agree with the reviewer's point to add more in-depth role of geoinformation in earthquake simulation. Spatial data and tools play a very important role in earthquake-stricken environments, such as assessing the extent of damage, investigating post-earthquake events (high-risk areas after the earthquake), examining the existing infrastructure in the region, and so on. In the simulation of urban environments, this role becomes more important and sensitive because we are faced with a large volume of important spatial data in decision making. Recommended tools generally have a spatial basis and use spatial data and analysis, even in earthquake engineering. To modify the text and emphasize this issue, the following sentences were added to the text.

In the agent-based USAR simulation section: "In previous studies, a geospatial information system (GIS) platform was used when preparing the environment and creating a simulation base map [1-3]. Spatial analysis and tools are used in most research endeavors in USAR operations [4]. Accurate and current spatial data play a vital role in earthquake-affected environments, such as assessing the extent

of damage and casualties, surveying vulnerable areas after an earthquake, examining the existing infrastructure in the region, and finally tremendous aid in humanitarian efforts [5, 6]. Risk assessment of urban areas limits the impact of harmful events by increasing awareness of their potential consequences using spatiotemporal data [7]. Recommended tools in earthquake engineering sciences generally have a spatial basis and use spatial data and analysis [7]. Basic information, maps, and spatial tools in the form of Spatial Decision Support Systems (SDSS) and spatial frameworks such as webGIS have a significant impact on the speed of USAR operations [8]. Earthquake environment simulation is one of the important parts of agent-based modeling which is implemented using spatial analysis. To evaluate the vulnerability of buildings, some models and software based on infrastructures' spatial parameters have been developed [2], such as U.S. Geological Survey (USGS) model [9], HAZUS-MH (Multi-Hazard) [10], JICA[1] model [3], Federal Emergency Management Agency (FEMA) fragility curves [11], PO-ZID[2], and PO-AB[3] parametric methods [12]) and CIPCast-ES (Critical Infrastructure Protection - Earthquake Simulator) simulator [8]."

3- **About the study area analysis, I ask the authors to pay more attention to the analysed bibliography. For example, to refer to a paper of 2009 about the probability of a new earthquake in the future when an earthquake actually occurred in 2017 is very strange to read. Moreover, I ask the authors to provide a reference for this statement about the North Tehran fault: 'It has the potential for a 7.2 magnitude earthquake'.**

Response:

Thank you for your in-depth analysis. According to various references, Iran has been introduced as one of the earthquake-prone areas and in recent years, several earthquakes have occurred in it. The authors themselves have witnessed a 6.2 magnitude earthquake in Tabriz and a 5.7 magnitude earthquake in Tehran. There is no doubt that Iran is seismic and according to previous studies, Iran is among the five seismic countries in the world [2]. In this regard, various studies have been conducted in the country by the International Institute of Earthquake Engineering and Seismology (IIEES) and the Crisis Management Organization of Iran, most of which are in Persian. The history of earthquakes in this region is presented in various articles, the following figures are examples of them [13].
* * *
[1] Japan International Cooperative Agency (JICA) model
[2] Potresne Odpornosti Zidanih (Slovenian language)
[3] Potresne Odpornosti armiranobetonskih konstrukcij (Slovenian language)

[Figure]

Figure 1- History of earthquakes around Tehran [13]

[Figure]

Figure 1- History of earthquakes around Tehran [14]

Alborz region, which is a mountain range located in the northern part of Tehran, witnessed a 7.3 magnitude earthquake in 1990 .In the city of Tehran, earthquakes greater than 7 have occurred throughout its history. The North Fault has also shown earthquakes between 6 and 7.2 in the past [2, 14, 15]. Therefore, a 7.2 magnitude earthquake is not an unlikely scenario. The values of the simulated earthquakes are different in various researches and a single number is not stated for the maximum earthquake potential. It should also be noted that the methodology presented in this research can be implemented for earthquakes with different magnitudes. In order to clarify the issue and remove the existing ambiguities, the paragraph was edited as follows:

In the case study and data section: "The recent Tehran earthquake (5.2 magnitude) in December 2017 attracted the attention of many urban planning organizations. Tehran is a highly seismic area because it is surrounded by the Ray, Masha-Fasham, and North Tehran faults (Figure 1(b)) [14]. Tehran is located in the southern part of the Alborz Mountains, where a magnitude 7.3 earthquake occurred in 1990 [16, 17]. Tehran faults show some M7+ historical earthquake records [2, 14]. Seismologists have reported that Tehran is vulnerable to earthquakes and is expecting a destructive earthquake in the future [3, 13]. The North Tehran fault (NTF) is the city's largest and most prominent active tectonic structure fault, which is approximately 175 km long [2, 18]. The paleoearthquakes study on this fault has revealed seven surface-rupturing events with magnitudes between 6.1 and 7.2 [2, 14, 15]. For this purpose, the North Tehran fault scenario, with the capacity to cause the most destructive potential earthquake in Tehran, was selected in the present study. The method developed in this research can be implemented for any scenario. In accordance with the previous earthquakes and suggestions of seismologist experts, we simulated 6.6, 6.9, and 7.2 magnitude earthquakes. The basic data used in environment simulation were block maps, population, distance from the fault, building material, agent location, year of building construction, and building height."

Thanks to the attention of the esteemed referee, "It has the potential for a 7.2 magnitude earthquake" sentence was edited as follows according to the history of the North Fault earthquakes:
In the case study and data section: "The paleoearthquakes study on this fault has revealed seven surface-rupturing events with magnitudes between 6.1 and 7.2 [2, 15]."

4- **About the material and methods section, the following issues merit attention. First, what does means this statement: 'We assume the presence of a disaster environment in which events are uncertain'? Moreover, about the simulation model, I would suggest the authors read the following two publications "Earthquake Simulation on Urban Areas: Improving Contingency Plans by Damage Assessment" and "A comprehensive system for semantic spatiotemporal assessment of risk in urban areas" published in the framework of an FP7 European project. i.e., CIPRNet, and that released a complex earthquake simulator (CIPCast-ES).**

Response:
We appreciate the reviewer's suggestions. The phrase "We assume the presence of a disaster environment in which events are uncertain" means that data and events in the earthquake environment are not definite and there is uncertainty in them. For example, it is not clear exactly how long it will take if we want to go from point A to point B. To clarify this sentence, the sentence was edited as follows:
In the scenario of the proposed agent-based USAR simulation section: "We assume the presence of a disaster environment in which events are uncertain, for example, the time it takes to go from location A to location B is not exactly known."

The two articles " Earthquake Simulation on Urban Areas: Improving Contingency Plans by Damage Assessment" and " A comprehensive system for semantic spatiotemporal assessment of risk in urban areas " were carefully studied. These two valuable articles will help us to use spatial data to simulate an earthquake-stricken environment and to provide a supportive decision-making system in the form of a WebGIS, which unfortunately we have not studied before. These two articles will be very useful in the field of environment simulation using the CIPCast-ES simulator and trying to present and implement the proposed method in the form of a webGIS. These two articles were used to improve the text of the article, also based on them ideas for future researches formed in the minds of the authors. Therefore, suggestions were made to readers to use CIPCast-ES to reduce the difficulty and

time required to simulate an earthquake-stricken environment. The highlights of these two articles, which helped document our research in different sections of the article, are as follows:

In the agent-based USAR simulation section: "To evaluate the vulnerability of buildings, some models and software based on infrastructures' spatial parameters have been developed [2], such as U.S. Geological Survey (USGS) model [9], HAZUS-MH (Multi-Hazard) [10], JICA[4] model [3], Federal Emergency Management Agency (FEMA) fragility curves [11], PO-ZID[5], and PO-AB[6] parametric methods [12]) and CIPCast-ES (Critical Infrastructure Protection - Earthquake Simulator) simulator [8]."

In the agent-based USAR simulation section: Risk assessment of urban areas limits the impact of harmful events by increasing awareness of their potential consequences using spatiotemporal data [7]. Recommended tools in earthquake engineering sciences generally have a spatial basis and use spatial data and analysis [7]. Basic information, maps, and spatial tools in the form of Spatial Decision Support Systems (SDSS) and spatial frameworks such as webGIS have a significant impact on the speed of USAR operations [8]."

In the conclusion section: "In this study, a simplified environment was considered, for example, it was assumed that the roads were not damaged. It is suggested that environment simulators such as Hazus or CIPCast-ES be used as a deterministic approach to simulate the damaged environment and the vulnerability of roads and. Future studies could also present the output of this study in the form of a spatial decision support system (SDSS) or a webGIS."

**5- The authors proposed the use of the VIKOR method. I could concur with the author that this method fits with the research aims. Indeed, in a research article, the choice of a methodology must be justified. Moreover, some description must be provided on this method. It is not sufficient to simply refer as follows "The interval-based VIKOR method has been previously described (Sayadi et al., 2009). Ordering is performed by the central agent." The authors must provide the required details.**

Response:
We generally agree with the reviewer's point to add detailed information. Some previous research has used formulas to calculate agent scores, including the following formula used by Rasekh and Vafaie Nejad ([14]).

$$\text{Cost} = X + (10 \times Y) + (10 \times Z) + (10 \times K) + T$$

X represents the time it takes for an agent to reach the activity position from his place, Y indicates the difficulty coefficient, Z represents the priority coefficient of activity performed by each agent, T represents the duration of each activity, and K represents the fatigue coefficient. All factors must follow the same unit. The use of such a formula is not very appropriate due to the fact that it strongly affects the results of the tender and is practically not an accepted formula and depends on experts' opinions. In this step, we use multi-criteria decision methods to evaluate the agents' bids. At this stage, all MCDM decision methods such as VIKOR, TOPSIS[7], ELECTRE[8], etc. can be used.
In previous researches, these methods have not been evaluated together in the field discussed in this article, and the predominant method in this field has not been recommended. On the other hand, since we use interval instead of a number in multi-criteria decision making, we could not use complex
* * *
[4] Japan International Cooperative Agency (JICA) model
[5] Potresne Odpornosti Zidanih (Slovenian language)
[6] Potresne Odpornosti armiranobetonskih konstrukcij (Slovenian language)
[7] Technique for Order of Preference by Similarity to Ideal Solution
[8] ÉLimination et Choix Traduisant la REalité

MCDM methods that could not consider an interval instead of a number. In previous studies, TOPSIS and VIKOR have been used based on interval values. In the article entitled "Developing an Agent-Based Simulation System for Post-Earthquake Operations in Uncertainty Conditions: A Proposed Method for Collaboration Among Agents" we used the interval-based TOPSIS method and in the article entitled " Agent-based task allocation under uncertainties in disaster environments: An approach to interval uncertainty" we used the interval-based VIKOR method to prioritize tasks. A review of the authors' previous research does not show any particular superiority or advantage between these methods. The interval-based VIKOR method is more considered in previous researches when the values are interval and it can be said that it is a more popular method. Therefore, in this study, we used the interval VIKOR method to evaluate and compare the agent's bid. In this research, the VIKOR method is used in two stages: ordering existing tasks and holding an auction. To determine the implementation stages of the VIKOR method, the following two sections were added to the text.

In the ordering existing tasks section: "At this stage, all MCDM decision methods such as VIKOR[9], TOPSIS[10], ELECTRE[11], etc. can be used. In previous researches, the predominant method in this field has not been recommended. On the other hand, since we use interval instead of a number in multi-criteria decision making, we use MCDM methods that can consider an interval instead of a number. In previous studies, TOPSIS and VIKOR have been used based on interval values, but no particular superiority has been observed [19-21]. In this research, the interval VIKOR method is used to sort tasks and compare the agents' bids. The VIKOR method is done in five main steps [21, 22]: First, the decision matrix with interval data is formed so that the rows represent the alternatives (A), the columns represent the criteria (C), and the matrix values ($f_{ij}$) represent the value of alternatives i relative to criterion j. Matrix values are interval ($f_{ij} \in \left[ f_{ij}^L, f_{ij}^U \right]$). Then the positive (PIS) and negative ideal solution (NIS) is determined. The positive ideal solution is the highest column value for the profit criterion and the lowest column value for the cost criterion. Then S and R intervals are calculated and based on them, the interval Q is calculated using the following formulas.

$$S_i^L = \sum_{j \in I} w_j \left( \frac{f_j^* - f_{ij}^U}{f_j^* - f_j^-} \right) + \sum_{j \in J} w_j \left( \frac{f_{ij}^L - f_j^*}{f_j^- - f_j^*} \right) \qquad i=1, ..., m$$

$$S_i^U = \sum_{j \in I} w_j \left( \frac{f_j^* - f_{ij}^L}{f_j^* - f_j^-} \right) + \sum_{j \in J} w_j \left( \frac{f_{ij}^U - f_j^*}{f_j^- - f_j^*} \right) \qquad i=1, ..., m$$

$$R_i^L = max \left\{ w_j \left( \frac{f_j^* - f_{ij}^U}{f_j^* - f_j^-} \right) \middle| j \in I, \quad w_j \left( \frac{f_{ij}^L - f_j^*}{f_j^- - f_j^*} \right) \middle| j \in J \right\} \qquad i=1, ..., m$$

$$R_i^U = max \left\{ w_j \left( \frac{f_j^* - f_{ij}^L}{f_j^* - f_j^-} \right) \middle| j \in I, \quad w_j \left( \frac{f_{ij}^U - f_j^*}{f_j^- - f_j^*} \right) \middle| j \in J \right\} \qquad i=1, ..., m$$

$$Q_i^L = v \frac{S_i^L - S^*}{S^- - S^*} + (1 - v) \frac{R_i^L - R^*}{R^- - R^*}$$

$$Q_i^U = v \frac{S_i^U - S^*}{S^- - S^*} + (1 - v) \frac{R_i^U - R^*}{R^- - R^*}$$

$$S^* = \min_i S_i^L, \qquad S^- = \max_i S_i^U, \qquad R^* = \min_i R_i^L, \qquad R^- = \max_i R_i^U$$

In the above formula, A* and A- are PIS and NIS, i is associated with benefit criteria, and J is associated with cost criteria, wj is the weight of criterion Cj and v is introduced as the weight of the strategy of ''the majority of criteria'' (or ''the maximum group utility''), here suppose that v = 0.5.
* * *
[9] Vlsekriterijumska Optimizacija I Kompromisno Resenje
[10] Technique for Order of Preference by Similarity to Ideal Solution
[11] ÉLimination et Choix Traduisant la REalité

Finally, find an appropriate alternative based on Q Intervals. A better option is to have a smaller Q interval than the others. The following constraints are used to calculate the smaller interval. If a=[$a^L$ $a^U$] and b=[$b^L$ $b^U$], the comparison between these two intervals is as follows:

- If the two intervals do not have an intersection, the interval whose values are lower is minimum interval number.
- If the values of the two intervals are $a^L \leq b^L < b^U \leq a^U$, the interval a is minimum if $\alpha(b^L - a^L) \geq (1-\alpha)(a^U - b^U)$.
- If the values of the two intervals are $a^L < b^L < a^U < b^U$, the interval a is minimum if $\alpha(b^L - a^L) \geq (1-\alpha)(b^U - a^U)$.

In the holding an auction section: "In some previous researches, special formulas have been used to calculate the agents' scores based on the difficulty of the task, the priority of the task, the duration of each activity, and so on [14]. The use of such a formula is not very appropriate due to the fact that it strongly affects the results of the action and is practically depends on expert opinions. In this step, instead of using the formula to evaluate the agents' bid, the interval VIKOR method is used to find better propose."

6- **About the results section, the authors have shown the Vulnerability maps for District 1. By the way, in the manuscript, they did not provide detail on how these maps were obtained. To specify the use of ArcGIS is not sufficient. The same thing about the use of JICA. First, the authors must specify this acronym, and then they must provide some detail about it to help the larger readability of this study.**

Response:
Thank you for pointing out this misunderstanding to us. To make the steps of preparing vulnerability maps more transparent using the JICA model in this research, the following figure and paragraphs were added in the methodology section. In this section, previous articles that have used this model were also referenced so that those interested can get acquainted with how to implement it by reading those articles in more detail.

In the USAR simulation section: "To simulate an earthquake-damaged environment, an earthquake risk assessment model was developed based upon the Japan International Cooperative Agency (JICA) model. The JICA model is the output of cooperation between the Center for Earthquake and Environmental Studies of Tehran and the JICA. The results of this project and its implementation have been presented previously [23] and used in various studies [3, 17, 19]. This model can calculate the buildings' level of destruction and the number of injured people based on the earthquake intensity, earthquake location, building vulnerability, and the population in these buildings [17]. The steps for creating a vulnerability map and finding casualties based on the JICA model are shown in Figure 2.

[Figure]

Figure 2. Building damage assessment: a) Steps for calculating the number of injured people, b) URML pre-code fragility curves [3, 6, 23].

In the results and discussion section, the following paragraph was added to clarify how to prepare a vulnerability map with the JICA model.

In the results and discussion section: "Based on the process presented in Figure 2 for different scenarios, the magnitude of the earthquake was calculated at the location of the buildings. Then, based on the vulnerability curve of buildings and the type of materials used, the amount of destruction of each building is determined."

In the results and discussion section: "After calculating the vulnerability of buildings and based on the formulas expressed in Figure 2, the numbers of injured and deceased people can be calculated using the JICA model."

7- **Moreover, also about the simulation, the authors specified in the results section that the AnyLogic software '… can process geospatial information system data'. Also, in this case, I suggest the authors provide more information about the environmental simulation in the materials and methods section rather than in the results section.**

Response:

Based on the suggestion of the esteemed referee, which is also a very useful opinion, the method of simulating the environment was transferred to the methodology section. Additional explanations on how to simulate the environment were also provided in the methodology section.

In the Materials and Methods section: "AnyLogic software is used to simulate the scenario described in the previous section based on multi-agent systems. In our scenario, we included four types of agents: injured individual, rescuer, coordinator, and central agent. In AnyLogic software, a statechart is designed to suit the tasks of each agent. Statechart determines the work process of the agent. The tasks described in the previous section were implemented for each agent. The simulated agents in the environment are independent, located in a specific place, logical and decision-making, can move to a specific location, and other agents, except the injured agent, communicate with each other. In simulating USAR operations, location and the use of maps and spatial analysis play a key role. AnyLogic software can process geospatial information system data. The initial locations of injured agents were based on building damage and the locations of rescue groups were randomly generated in the environment. The definitions of agents and their characteristics were described in detail in our previous article [19]. The relevant agents move along the central line of the road and use the Dijkstra algorithm to find the shortest path. Dijkstra's algorithm is a well-known algorithm for finding the shortest paths in road networks [4].

There are many injuries in the environment. The rescue of the injuries is possible with the cooperation of the agents. The process of cooperation between agents is simply put as follows: The central agent first sorts the tasks according to their priorities. After the coordinating agent has been determined, the central agent sends the task properties to the coordinating agent. The coordinator holds an auction. Rescue agents bid in accordance with their environmental and working conditions. Rescuers are in a ready state at the start of the operation. Each successful rescue agent moves to the task's location. After reaching the task position, the rescue agent begins rescuing the injured agents. During the execution of their assigned work, the agents may find considerable differences between the real-world information and the information expressed in the auction. In such instances, the agents may stop performing their tasks and report the discrepancies to the central agent. The method of cooperation between the agents is described based on the proposed method in the next section."

8- **From my side, it is a bit strange to read a scientific article without any discussion section. Indeed, some discussion is provided in the results section, but it is not sufficient and must be improved. In doing that, to refer to other research experiences in comparing this one is expected. In other words, it is not acceptable to have a scientific article without proposer discussions.**

Response:

The authors completely agree with the reviewer's comment and have revised the section as follows to make the results understandable. The error occurred in expressing the title of the "Results" section, and the title was changed to "Results and Discussion". In order to improve the article, the discussion regarding the output of the vulnerability of buildings, the number of injured people, and the discussion regarding the output of the proposed task allocation method were added to the text of the

article. At this stage, the output of the present study was compared with the output of previous researches in each stage. In this regard, the following paragraphs were added to the text.

In the results and discussion section: "The results of estimating the vulnerability of District 1 for scenario 6.6 show the complete destruction of 18% of the buildings (7,063 buildings of all buildings) in the study area, which is mostly located in the central and northeast part of the region. For this scenario, 27% of the extensive destruction is observed in the buildings, so that these buildings are uninhabitable and there is a possibility of the vulnerability of people in this category of buildings. In the 6.9 magnitude scenario, 29% of complete destruction and 31% of extensive destruction is observed in buildings. It is obvious that with the increase of earthquake intensity, the amount of destruction of buildings increases. Scenario 7.2 shows that with this intensity, 53% of the buildings are severely damaged and these buildings will not be usable. Most of the damaged buildings are located in the central part of the region. A similar result has been obtained in research [2], the main reason being the high structural density and population in this part of District 1. In previous researches, Hashemi and Alesheikh (2011) estimated the number of complete damaged buildings 4% and 32% damage for District 10 of Tehran based on the 6.4 Richter for the Mosha Fault. Hooshagi and Alesheikh (2018) estimated the damage to buildings for the city of Tehran at 16% complete destruction and 24% extensive destruction based on the 6.6 Richter scenario in Niavaran Fault. The degree of degradation of 18%, 29% and 53% according to scenarios 6.6, 6.9 and 7.2 in this study is almost similar to previous researches."

In the results and discussion section: "According to Table 2, as the magnitude of the earthquake increases, the number of people dead and injured increases, so that in the 7.2 magnitude earthquake, 58% of the people were directly involved. Based on the JICA model, Mansouri et al. (2008) estimated the number of deaths and injuries related to the 6.7 magnitude Riches earthquake at 7% and 4%, respectively. The percentage of people who died and were injured in their research is similar to the 6.6 magnitude earthquake scenario in our research."

In the results and discussion section: "Using the proposed strategies, the smallest improvement in results with uncertainty was 2.9 h (13%) for a scenario with 2000 agents and 28,856 tasks (6.6 magnitude earthquake). The maximum improvement was 60.6 h (21%) hours for 1000 agents and 111,463 tasks. In a previous study, task allocation progress was 18% when uncertainty was applied in a laboratory environment regardless of spatial strategies [20]. Previous research has also shown that that taking uncertainty in task allocation into account in District 3 of Tehran improved the duration of the rescue operations by 20%, and decreased the number of fatalities by 15% [19]. Therefore, the proposed approach in this study showed a better performance than the traditional CNP methods."

In the results and discussion section: "The reduction rate ranged from 54% to 60% when the number of agents was doubled. The duration of a USAR operation increased when the number of tasks increased for a given number of agents. Therefore, the duration of the rescue operation was related to the number of rescue agents and the number of available tasks in a scenario. There was an inverse relationship between the duration of the USAR operation and the number of rescue agents, and a direct relationship between the duration of the operation and the number of tasks."

9- **In the conclusion section, at least one or two statements on the limits of the proposed methods should be expected.**
Response:
Thank you for your in-depth analysis and insightful comments. The main limitations of the proposed method are two cases, which were added as follows to clarify the issue:

In the conclusion section: "One of the limitations of agent-based simulation is the difficulty of implementation and time-consuming processes that require the availability of powerful processors. Although the proposed method is simple in terms of interval uncertainty and does not perform complex calculations, it is time-consuming due to a large number of calculations to apply spatial strategies for each task and each agent. These processes require the availability of powerful processors. Another limitation of the proposed method is the assumption of communication between agents and the central agent. The proposed method is developed for assistant agents in which groups' information (as agents) is transmitted between groups through tools such as mobile phones. Although the volume of message transfer in this method is less than the traditional CNP method, in severe earthquakes that damage the internet infrastructure, the task allocation method still has problems. Of course, this limitation exists in all communication methods that consider the whole groups."

**Technical comments**

1- **Captions of figures and tables must be self-explicative. For example, in figure 1, a map of PGA is reported but this acronym and its significance is not explained in the caption.**

Response:

The authors completely agree with the reviewer's comment and have revised all captions of images and tables. The sentences were reviewed as follows to make the figures and tables understandable.

[Figure]

**Figure 1** Location of case study: (a) peak ground acceleration (PGA) map of Iran for a return period of 2475 years and approximate location of Tehran, (b) location of District 1 and active faults in Tehran (c) Map of District 1 (study area) and active faults, Tehran.

**2- Case study: population of the study area referred to which year?**

Response:

Thanks to this statement. The population census in Iran is conducted every 5 years. The statement is related to the last census conducted in 2016, which was mentioned in the text of the article to clarify the issue.

In the case study and data section: "According to the 2016 census, its population is 433,500 people."

We believe that our manuscript is substantially improved and has no similarity to our previous articles. We would be glad to respond to any further questions and comments that you may have. Yours Sincerely

[revised manuscript text omitted]